# Insect herbivory dampens Subarctic birch forest C sink response to warming

Tarja Silfver [1,2✉], Lauri Heiskanen [3], Mika Aurela [3], Kristiina Myller [4], Kristiina Karhu[5], Nele Meyer [5,6], Juha-Pekka Tuovinen [3], Elina Oksanen[4], Matti Rousi[7] & Juha Mikola [1,2]

Climate warming is anticipated to make high latitude ecosystems stronger C sinks through increasing plant production. This effect might, however, be dampened by insect herbivores whose damage to plants at their background, non-outbreak densities may more than double under climate warming. Here, using an open-air warming experiment among Subarctic birch forest field layer vegetation, supplemented with birch plantlets, we show that a 2.3 °C air and 1.2 °C soil temperature increase can advance the growing season by 1–4 days, enhance soil N availability, leaf chlorophyll concentrations and plant growth up to 400%, 160% and 50% respectively, and lead up to 122% greater ecosystem $CO_2$ uptake potential. However, comparable positive effects are also found when insect herbivory is reduced, and the effect of warming on C sink potential is intensified under reduced herbivory. Our results confirm the expected warming-induced increase in high latitude plant growth and $CO_2$ uptake, but also reveal that herbivorous insects may significantly dampen the strengthening of the $CO_2$ sink under climate warming.

[1] Faculty of Biological and Environmental Sciences, University of Helsinki, Niemenkatu 73, 15140 Lahti, Finland. [2] Kevo Subarctic Research Institute, Biodiversity Unit of the University of Turku, Kevontie 470, 99980 Utsjoki, Finland. [3] Finnish Meteorological Institute, Climate System Research, P.O. Box 503, 00101 Helsinki, Finland. [4] Department of Environmental and Biological Sciences, University of Eastern Finland, P.O. Box 111, 80101 Joensuu, Finland. [5] Faculty of Agriculture and Forestry, University of Helsinki, P.O. Box 27, 00014 Helsinki, Finland. [6] Department of Soil Ecology, University of Bayreuth, Dr.-Hans-Frisch-Straße 1-3, 95448 Bayreuth, Germany. [7] Natural Resources Institute Finland, Latokartanonkaari 9, 00790 Helsinki, Finland. ✉email: tarja.h. silfver@gmail.com

Temperature is one of the key factors that control $CO_2$ exchange between the ecosystem and the atmosphere. In high latitude ecosystems, climate warming directly stimulates plant production by providing a warmer environment and longer growing season for $CO_2$ fixation[1,2], but also indirectly by accelerating decomposition and nutrient release for plant uptake[3,4]. If plant growth is enhanced more than decomposition, ecosystems become stronger C sinks[5]. Avian[6–9] and mammalian[10,11] grazers are known to significantly suppress gross primary production (GPP) in Arctic ecosystems by consuming plant photosynthetic tissues and changing plant community structure. Grazing can limit the positive response of plant production and $CO_2$ uptake to climate warming in grazed areas[7,10], although the ecosystems in general may be greening[12,13] and their growing season C uptake increasing[13,14]. Recent evidence further suggests that when occurring at outbreak densities, herbivorous insects can dramatically suppress GPP in the Arctic[15]. Herbivorous insects are ubiquitous however, and even at their background, non-outbreak densities, can annually consume 1–15% of plant foliage[16–18] and contribute to nutrient cycling in ecosystems[19]. Minor chronic herbivory can reduce plant growth[20,21] and in the long-term, background herbivory may have a stronger effect on woody plant growth than the devastating, short-term outbreaks[22,23]. However, while there are studies of the effects of insect herbivory on leaf chlorophyll fluorescence and

photosynthesis rates at high latitudes[22,24], no study has yet examined how background insect herbivory influences ecosystem–atmosphere $CO_2$ exchange in these areas.

Both paleontological data from past intervals of significant climate change[25] and observations along latitudinal[16,26–29] and elevational[30] gradients indicate increasing insect herbivory with warming. The positive association between the warmest summer month temperature and the level of herbivore damage seems to be particularly strong in cold-limited, high latitude ecosystems[16,26–29]. Here, we provide evidence that background insect herbivory can significantly weaken the $CO_2$ uptake potential of a Subarctic mountain birch forest ecosystem and offset the strengthening of C sink under climate warming (Fig. 1). Our results are from an open-air warming experiment, established at the Kevo Subarctic Research Institute, North Finland (69°45.4′N, 27°00.5′E) in 2016 when 20 plots ($1 \times 0.75$ m$^2$), each containing intact field layer vegetation but no adult trees, were created. Twelve *Betula* plantlets (including local *Betula nana* and *Betula pubescens* subsp. *czerepanovii* and currently slightly more southern *Betula pendula* and *B. pubescens*) were planted within field layer vegetation in each plot to include Subarctic woody species in the experiment and to have controlled and well replicated plant material to accurately follow treatment responses. During the periods of warming (May–November), green metal plates (mimicking plant leaves) were heated to

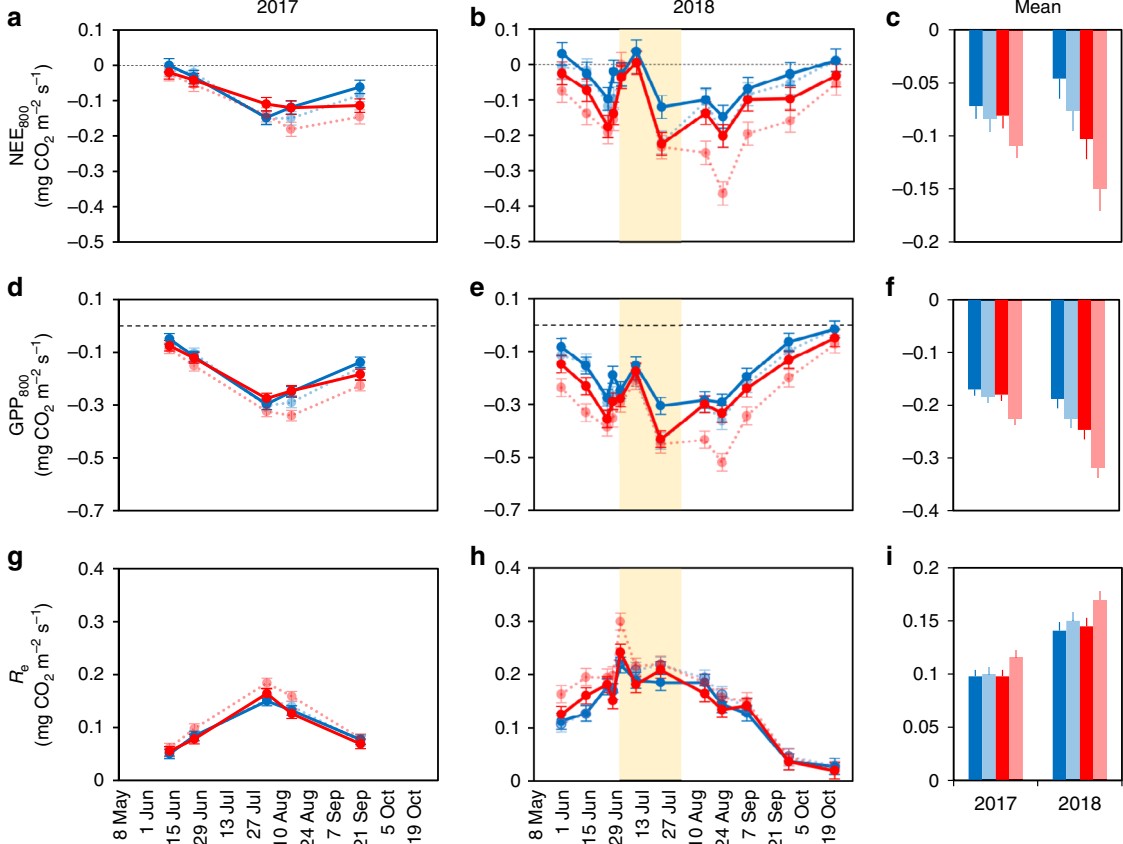

**Fig. 1 Warming and reduction in herbivory increase GPP and ecosystem $CO_2$ uptake potential.** Left hand and middle panels show the variation in daytime $CO_2$ fluxes: **a, b** net ecosystem exchange NEE$_{800}$, **d, e** gross primary production GPP$_{800}$, and **g, h** ecosystem respiration $R_e$ in warmed (red) and ambient (blue) field plots of Subarctic mountain birch forest field layer vegetation (supplemented with cloned birch plantlets) during growing seasons 2017 and 2018 (dots are estimated marginal means ± s.e.m. produced by the fitted statistical model in Table 2; $n = 5$ field plots). Right hand panels show seasonal means ± s.e.m. of **c** NEE$_{800}$, **f** GPP$_{800}$, and **i** $R_e$ (means are estimated marginal means produced by the fitted statistical model in Table 2, $n = 5$ field plots examined over 5 [year 2017] or 12 [year 2018] repeated measures). Light tone dashed lines, symbols, and bars denote plots, where herbivory was reduced using an insecticide. The pale yellow background stripe stands for the period of severe hydrological stress (2nd July–1st August, 2018; see Fig. 3). Source data are provided as a Source Data file.

approximately 3.3 °C above ambient temperature in half of the plots, using infrared ceramic heaters. Heating was controlled by real-time, replicated temperature measurements in the control and heated plots, and led to approximately 2.3 °C warmer moving air and 1.2 °C warmer soil in the heated plots. Following a fully factorial 2 × 2 set-up (two levels of warming and two levels of herbivory yielding four warming × herbivory combinations with $n = 5$ for each), half of the plots were also sprayed weekly using an insecticide to reduce insect herbivory. We measured the ecosystem–atmosphere $CO_2$ fluxes in each plot with static chambers through two remarkably different growing seasons, and to explain the observed patterns in fluxes, we surveyed: the phenology, shoot growth, leaf chlorophyll content, and damage of *Betula* plantlets; soil microbial biomass and mineral N availability; and abiotic attributes such as air and soil temperature and moisture. Our results reveal that warming increases the $CO_2$ uptake potential in Subarctic ecosystems, but also that the generally minor background insect herbivory can strikingly control the $CO_2$ exchange, in both present and future climates. Thus, the $CO_2$ uptake of high-latitude ecosystems in the future likely depends not only on the magnitude of temperature rise, but also on the levels of insect herbivory.

## Results and discussion

**Warming, insect herbivory, and $CO_2$ fluxes**. At high latitudes, the level of background insect herbivory is typically low[26,29]. In accordance with this, the herbivore damage of our experimental *Betula* plantlets was only 2–20% of the damage earlier observed ca. 1000 km south of our site[20,21]. On average, 26% of all *Betula* leaves in our control plots with natural herbivory were wounded by insects in 2017 (Supplementary Table 1). The observed damage index was low (Fig. 2d) as most of the damaged leaves had only 1–4% of their leaf area damaged (Supplementary Table 1). This finding corresponds well with earlier estimates of 1–2% of *Betula* leaf area damaged due to background herbivory at high latitudes[26,29]. However, despite such a low level of damage, the 67% reduction in mean leaf damage index (Fig. 2d and Table 1), achieved by the insecticide treatment in our study, had a remarkably strong effect on the ecosystem C uptake potential. The estimates of daytime $CO_2$ fluxes showed that reducing leaf damage by two-thirds increased the ecosystem C sink potential by, on average, 26% and 52% in 2017 and 2018, respectively (i.e., the potential net ecosystem exchange [$NEE_{800}$] of $CO_2$ was more negative with reduced herbivory; Fig. 1a–c and Table 2). These values are similar to the 22 and 107% increase in C sink potential due to the experimental warming

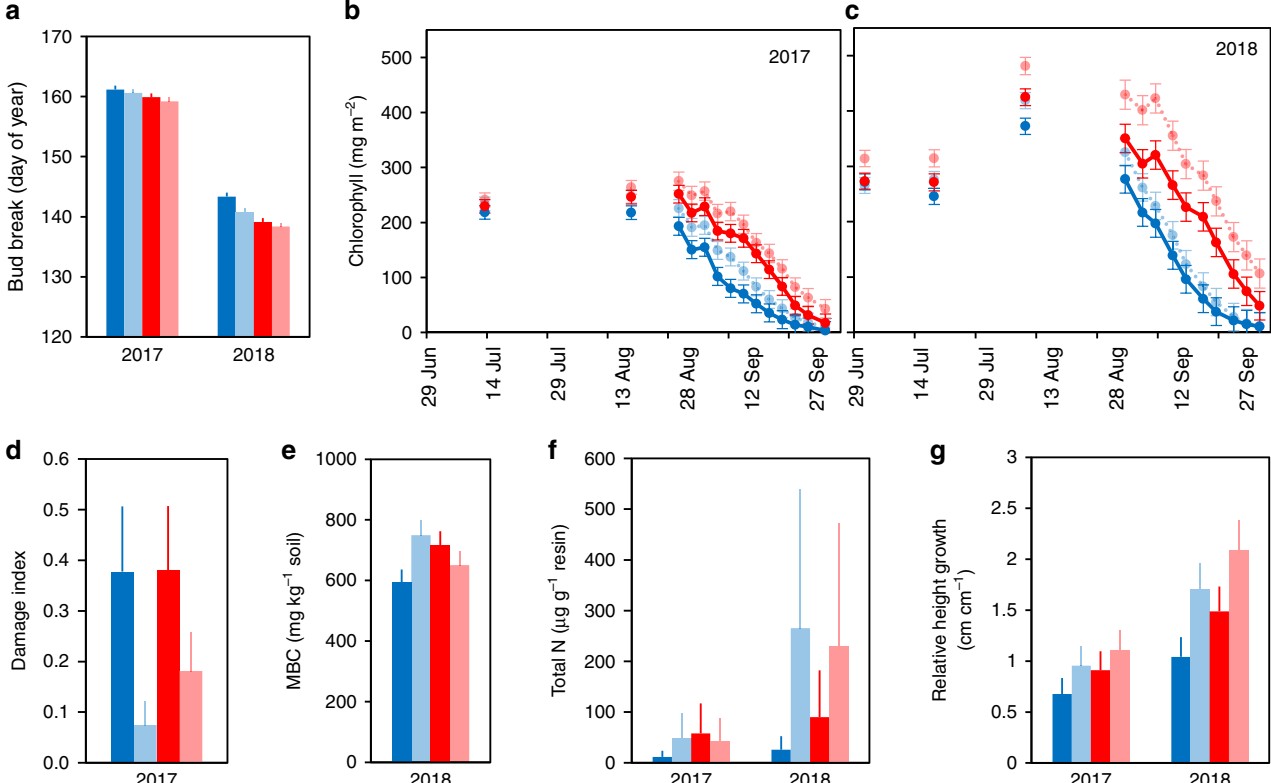

**Fig. 2 Longer growing seasons and enhanced soil mineral N explain better plant growth under warming and reduced herbivory.** Birch plantlets had **a** earlier spring bud break (Supplementary Table 3; for each mean $n = 57$–60 plantlets growing in 5 replicate field plots) and **b**, **c** higher leaf chlorophyll content in summer and autumn (Supplementary Table 4; for each dot in 2017 $n = 57$–60 and for each dot in 2018 $n = 39$–60 plantlets growing in 5 replicate field plots) in plots with warming (red symbols) and reduced herbivory (light tone symbols), compared to plots with ambient temperature (blue symbols) and natural herbivory (dark tone symbols), which responses together resulted in longer growing seasons. As warming did not increase **d** leaf herbivore damage (Table 1; for each mean $n = 57$–60 plantlets growing in 5 replicate field plots), these effects together with greater **e** soil microbial biomass carbon, MBC (Supplementary Table 5; $n = 5$ field plots, each examined at 3 depths of soil) and **f** resin bag mineral N ($NH_4^+$ and $NO_3^-$) capture (Supplementary Table 5; $n = 5$ field plots) under warming and reduced herbivory, apparently explain the increased **g** relative growth of birch plantlets (Supplementary Table 3; for each mean $n = 57$–60 plantlets growing in 5 replicate field plots) under these treatments, which in turn likely explains the patterns observed in the $CO_2$ exchange of the entire field layer vegetation (Table 2 and Fig. 1). All means are estimated marginal means (±s.e.m.) produced by the fitted statistical models in Table 1 and Supplementary Tables 3–5. Means and errors for MBC were back-transformed from square root-transformed data and those for leaf damage index, plant growth, and total soil mineral N from log-transformed data. Source data are provided as a Source Data file.

(Fig. 1a–c and Table 2) and provide, to our knowledge, the first evidence that the seemingly low background herbivory can significantly affect $CO_2$ uptake in Subarctic ecosystems.

Warming and herbivory effects on $NEE_{800}$ increased with time: the warming effect was more pronounced in the late rather than the early growing season in 2017 (significant date × warming interaction; Table 2 and Fig. 1a), and in 2018, warming increased ecosystem C sink throughout the season, except during the heat wave (Figs. 1b and 3b). Increased treatment effects likely originate from the cumulative size differences between the plants in control and treatment plots (Fig. 2g). Supporting earlier studies with avian[7,8] and mammalian[10,11] herbivores, we found that shifts in $NEE_{800}$ were mainly driven by shifts in $GPP_{800}$ (Fig. 1a–f), although ecosystem respiration ($R_e$) was also affected by the treatments (Table 2 and Fig. 1g–i).

Earlier observations in Fennoscandia show that *B. pubescens* foliar damage increases from 1–2 to 5–7% of leaf area eaten along a latitudinal gradient from 70°N to 60°N[26]. This gradient climatically roughly equals the predicted warming in the next 100 years[31]. A similar gradient of increasing leaf damage has also been found for *B. pendula*[26], and together these observations suggest that leaf damage may more than double in northern birch forests during the expected warming. Mathematical simulations produce similar trends, but with a wide range of magnitude: for a *Betula glandulosa–nana* complex a 20% increase in leaf damage is predicted along a 3 °C increase in temperature in the Arctic[29] and for *B. pubescens* a 30–450% increase along a 1.7 °C increase in Scandes and a 50–200% increase along a 3.5 °C increase in Northern Russia[32]. Based on these observations and predictions, we expected higher leaf damage in our warmed plots, but warming did not increase leaf damage (Fig. 2d), except for *B. pubescens* under reduced herbivory (a significant warming × herbivore reduction × *Betula* species interaction effect in Table 1). Even this interaction effect is likely coincidental, since we found no main or interaction effects of warming on leaf damage when we repeated the damage survey in 2019 (Supplementary Table 2).

Although our results seem to suggest that earlier predictions of increasing herbivory with warmer climate[16,25–30,32] would not hold when tested experimentally, this interpretation needs to be treated cautiously as treatment plots in field experiments are vulnerable to both congregation and avoidance of freely moving animals and may therefore tell little about the responses of herbivore abundances under large-scale climatic changes[33]. While

**Table 1 Statistics of warming and herbivory effects on leaf damage index.**

|  | F | P |
|---|---|---|
| Soil OM content | 0.3 | 0.563 |
| Vascular plant cover | 0.0 | 0.970 |
| Lichen cover | 2.8 | 0.096 |
| Moss cover | 0.8 | 0.362 |
| Warming (W) | 3.1 | 0.079 |
| Herbivory reduction (H) | 42.7 | **<0.001** |
| *Betula* species (S) | 5.6 | **0.023** |
| W × H | 3.5 | 0.062 |
| W × S | 0.8 | 0.471 |
| H × S | 0.5 | 0.662 |
| W × H × S | 3.5 | **0.016** |

The data collected in 2017 was log-transformed and analyzed using linear mixed models and Type I ANOVA (with two-sided significance tests). OM content and the areal cover of vascular plants, mosses, and lichens were treated as covariates and added to models to remove plot-to-plot variation that might otherwise confound the treatment effects. Field replicate block and birch genotype (nested within species) were included in the models as random effects, but are not reported. $N = 234$ experimental birch plantlet. F and P indicate F-statistics and P-values respectively; $P < 0.05$ are in bold.

**Table 2 Statistics of warming and herbivory effects on $NEE_{800}$, $GPP_{800}$, and $R_e$ in 2017–2018.**

|  | $NEE_{800}$ F | $NEE_{800}$ P | $GPP_{800}$ F | $GPP_{800}$ P | $R_e$ F | $R_e$ P |
|---|---|---|---|---|---|---|
| **2017** |  |  |  |  |  |  |
| Soil OM content | 0.2 | 0.687 | 6.8 | **0.026** | 41.0 | **<0.001** |
| Vascular plant cover | 9.3 | **0.014** | 29.2 | **<0.001** | 42.6 | **<0.001** |
| Lichen cover | 20.5 | **0.001** | 8.4 | **0.015** | 27.4 | **0.001** |
| Moss cover | 1.8 | 0.211 | 0.1 | 0.712 | 7.6 | **0.024** |
| Date (D) | 46.5 | **<0.001** | 121 | **<0.001** | 136 | **<0.001** |
| Warming (W) | 2.0 | 0.190 | 4.6 | 0.062 | 3.1 | 0.119 |
| Herbivory reduction (H) | 5.7 | **0.043** | 13.2 | **0.006** | 11.6 | **0.010** |
| W × H | 1.0 | 0.352 | 3.5 | 0.094 | 7.0 | **0.030** |
| W × D | 2.9 | **0.029** | 1.3 | 0.282 | 1.3 | 0.286 |
| H × D | 1.1 | 0.355 | 1.6 | 0.188 | 0.8 | 0.541 |
| W × H × D | 0.2 | 0.916 | 0.5 | 0.766 | 0.3 | 0.891 |
| **2018** |  |  |  |  |  |  |
| Soil OM content | 0.02 | 0.895 | 2.4 | 0.153 | 10.2 | **0.011** |
| Vascular plant cover | 0.7 | 0.411 | 8.8 | **0.014** | 26.6 | **0.001** |
| Lichen cover | 8.2 | **0.016** | 3.5 | 0.085 | 2.0 | 0.185 |
| Moss cover | 5.2 | **0.046** | 7.6 | **0.018** | 2.8 | 0.125 |
| Date (D) | 31.5 | **<0.001** | 60.8 | **<0.001** | 87.4 | **<0.001** |
| Warming (W) | 13.2 | **0.006** | 13.9 | **0.004** | 1.6 | 0.239 |
| Herbivory reduction (H) | 8.3 | **0.020** | 13.1 | **0.006** | 7.3 | **0.033** |
| W × H | 0.4 | 0.534 | 1.3 | 0.280 | 1.7 | 0.238 |
| W × D | 1.5 | 0.129 | 1.2 | 0.264 | 2.4 | **0.008** |
| H × D | 1.3 | 0.215 | 1.3 | 0.223 | 0.4 | 0.944 |
| W × H × D | 1.5 | 0.154 | 1.4 | 0.158 | 0.7 | 0.775 |

Treatment effects on net ecosystem exchange ($NEE_{800}$), gross primary production ($GPP_{800}$), and ecosystem respiration ($R_e$) were tested using repeated measures linear mixed models and Type I ANOVA (with two-sided significance tests), where the variance is allocated to explanatory variables in the order of their appearance. Soil organic matter (OM) content and cover of vascular plants, lichens, and mosses are continuous variables that describe the variation among the experimental plots prior to the establishment of the experiment. They were used in the models as covariates to remove plot-to-plot variation that might otherwise confound the treatment effects. Years were analyzed separately ($N = 100$ for 2017, $N = 240$ for 2018). Date was treated as a repeated measure, warming and herbivory reduction as fixed effects, and treatment block (not reported) as a random effect. F and P indicate F-statistics and P-values respectively; $P < 0.05$ are in bold.

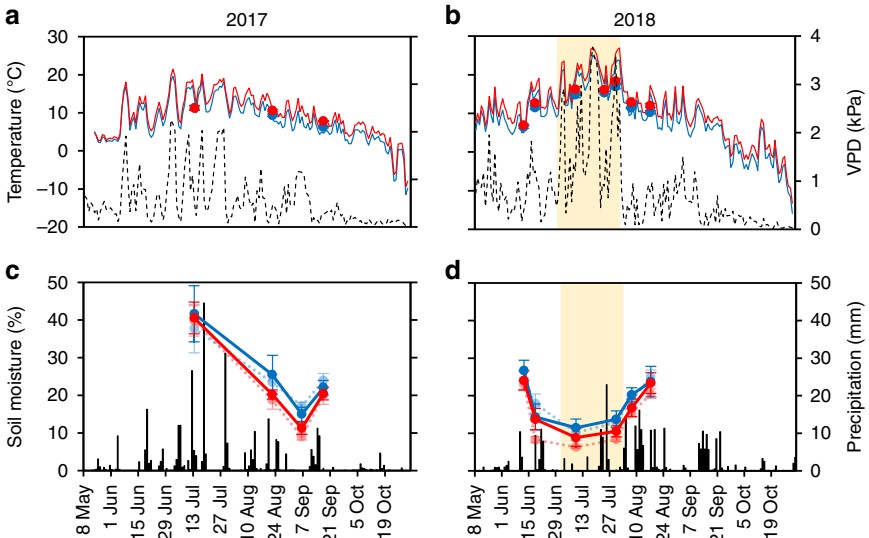

**Fig. 3 Abiotic attributes during periods of warming in 2017 and 2018. a, b** Air temperature (solid lines are mean daily temperatures of warmed [red] and ambient [blue] plots, $n = 3$ field plots), soil temperature (dots are mean ± s.e.m. of warmed [red] and ambient [blue] plots, $n = 10$ field plots, each examined through 3–5 within-plot measurements) and daily maximum of atmospheric water vapor pressure deficit (VPD; dashed black line); **c, d** soil moisture (dots are means ± s.e.m. of warmed [red] and ambient [blue] plots, light tone dashed lines and symbols denote plots, where herbivory was reduced using insecticide; $n = 5$ field plots, each examined through 3–5 within-plot measurements) and daily precipitation (thin black bars). Precipitation and VPD data are from the Kevo weather station (Finnish Meteorological Institute), located 200 m from the experimental site. The pale yellow background stripe stands for the period of severe hydrological stress (2nd July–1st August, 2018). Source data are provided as a Source Data file.

this bias precludes reliable predictions of warming effects on future herbivory in our and other plot-level experiments[33], experimental herbivory treatments, as the one in our study, can be used to predict and disentangle the effects of future changes in herbivore pressure on $CO_2$ fluxes from those deriving directly from increasing temperatures.

**Mechanisms behind the changes in $CO_2$ fluxes.** Our results reveal how warming and reduction in herbivory affected many of those plant traits and ecosystem properties that can control ecosystem C sink capacity through their effects on plant photosynthesis and growth. We found 1–4 days earlier bud break (Fig. 2a and Supplementary Table 3) and significantly later autumnal decline in leaf chlorophyll content in the warmed compared to control *Betula* plantlets (a significant date × warming interaction effect for both years; Supplementary Table 4 and Fig. 2b, c). This was expected, and accords with the satellite observations of prolonged growing seasons in high latitude ecosystems during the recent decades of warming[2]. More surprisingly, we also found a bilateral extension of the growing season in plantlets that had reduced herbivore load (Fig. 2a–c and Supplementary Tables 3 and 4). These results show that both warming and insect herbivory can have an impact on the length of the period when plants are able to grow and fix $CO_2$ from the atmosphere. Considering the key role of N deficiency in limiting primary production at high latitudes[4], another major finding from our study is that warming and herbivore reduction, alone and in combination, increased the availability of mineral N (sum of $NH_4^+$ and $NO_3^-$) in the soil by nearly 8-fold by the end of the second, full, growing season (Fig. 2f). Unexpectedly, reduced herbivory had a stronger effect on N mineralization than warming, and warming increased N availability under natural herbivory only (a significant warming × herbivore reduction interaction effect; Supplementary Table 5).

Warmer soil (Fig. 3a, b) can, as such, enhance microbial activity and nutrient mineralization[34], but why would plant release from herbivory also lead to higher soil N availability? We propose here that the observed positive effects of both warming and herbivory reduction were, for the most part, linked to the 30% and 50% increase in plant growth, respectively, under these treatments (Fig. 2g). Recent evidence from a subalpine forest suggests that in N-limited environments, trees can sustain greater growth under warming by increasing fine root production and release of root exudates, which stimulate microbial activity and N mineralization in the soil[35]. As greater shoot growth is likely associated with higher belowground C allocation, such a feedback loop through priming of organic matter decomposition could logically explain our findings of improved mineral N availability in the plots with reduced herbivory and enhanced plant growth. Supporting this interpretation, we found greater microbial biomass in the soil in response to both warming and reduced herbivory (Fig. 2e and Supplementary Table 5). While these effects were not fully additive as microbial biomass and N availability were not highest in the combined warming-herbivory reduction treatment with the greatest plant growth (Fig. 2g), it is likely that the improved N availability in the soil led to higher chlorophyll content in summer leaves (Fig. 2b, c and Supplementary Table 4), which in turn enhanced the ecosystem C sink through greater $GPP_{800}$ under warming and reduced herbivory (Fig. 1a–f).

Our results add to earlier findings, which suggest that plant growth in the Arctic is heavily controlled by low air and soil temperatures. For instance, in a study with comparable shrub dominated field layer vegetation, *B. glandulosa* had a 2.5-fold growth increase with 2.5 °C warmer air and 2.3 °C warmer soil[36]. Our study provides evidence that plant growth and C sink potential also seem to be heavily controlled by background insect herbivory. The availability of mineral N was negligible in our control plots (Fig. 2f), suggesting that plants at our site should be unable to compensate for losses in leaf mass. This was exactly the case; despite the low level of herbivory, the growth of *Betula* plantlets was remarkably slower under natural than reduced herbivory (Fig. 2g). One potential explanation for the extensive growth and GPP responses to herbivory reduction is that the leaf

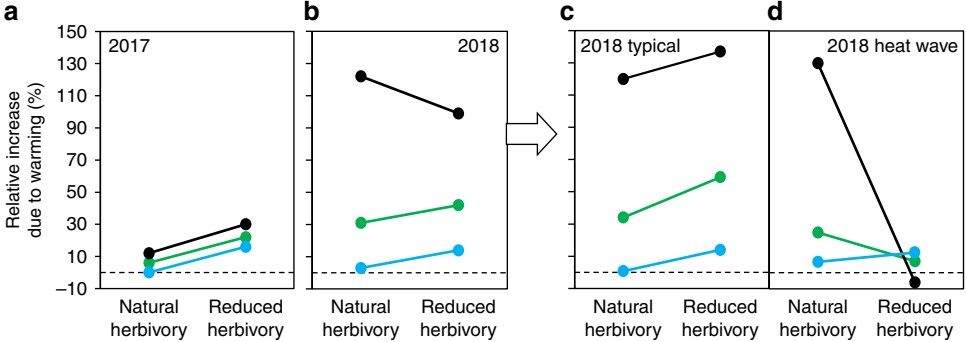

**Fig. 4 Schematic illustrations of how herbivory modified the effects of warming on ecosystem $CO_2$ exchange.** Year 2017 responses are shown as **a** overall means only while year 2018 responses are shown as both **b** overall means and split into two periods: **c** typical weather conditions and **d** a period of an extreme heat wave (pale yellow stripe in Figs. 1 and 3). The relative increase in gross primary production $GPP_{800}$ (green dots), net ecosystem exchange $NEE_{800}$ (black dots), and ecosystem respiration $R_e$ (blue dots) due to warming was higher when herbivore pressure was reduced in 2017 and in 2018 under typical weather conditions, but not during the 2018 heat wave. These responses suggest that plants with reduced herbivory benefited more from warming (in terms of $CO_2$ fixing) than plants with natural herbivory under typical humid conditions, but not under the heat wave, apparently because of being more susceptible to hydrological stress due to greater size (Fig. 2g) and higher water consumption (Fig. 3d).

area where photosynthesis is deleteriously affected by folivory can be 6-fold in comparison to the actual consumed area[37,38], possibly due to suppression of the efficiency of photosystem II in the remaining leaf tissue[39]. In northern deciduous trees, herbivory also commonly induces production of defence compounds such as phenolics[40] and when herbivory is reduced, resources allocated to induced defence are likely to be used for growth. Reducing herbivory may also simultaneously reduce leaf pathogens[41,42], leading to a more positive effect on GPP than could be predicted by purely measuring the reduction in herbivory damaged leaf area.

**Insect herbivory impact under typical weather and heat wave.** In line with the 50% increase in plant growth with herbivory reduction (Fig. 2g), we found that the relative increase in $GPP_{800}$, $NEE_{800}$, and $R_e$ due to warming was higher when herbivore pressure was reduced (Fig. 4a), although statistically significant warming × herbivory interaction effects were observed for $R_e$ only (Table 2 and Fig. 1g). In 2017, warming elevated $GPP_{800}$ and $NEE_{800}$ by, on average, 22% and 30%, respectively, when herbivory was reduced, but only by 6% and 12% under natural herbivory (Fig. 4a). These results suggest that plants were relatively better able to utilize the benefits of warming when insect herbivory was reduced, and as a result, the ecosystem acted as a stronger C sink. A similar pattern was not, however, observed for $NEE_{800}$ in 2018, but instead, the mean relative increase in $NEE_{800}$ was higher under natural herbivory (Fig. 4b). This change in the pattern seems to be related to differences in weather conditions between 2017 and 2018.

There was a month-long period of intense heat wave during the peak growing season in 2018. In July, the mean temperature was 5 °C higher and the total precipitation 10% lower than the long-term averages (Supplementary Table 6), soil moisture was low (Fig. 3d), and the daily maximum atmospheric water vapor pressure deficit (VPD) was frequently above 2.5 kPa (Fig. 3b), indicating severe moisture stress[43]. Of all our *Betula* plantlets, 19% died during the warmest days (July 18–20) despite irrigation (the amount of water given to plots each of the 3 nights corresponded to precipitation during a heavy thunderstorm). Notably, the drought was more extreme in plots with warming and reduced herbivory as shown by the lower soil moisture under these treatments (Fig. 3d; a marginally significant [$P = 0.054$] date × warming × herbivory reduction interaction effect in Supplementary Table 7), which likely resulted from the larger size and higher transpiration of plants in these plots.

As plant photosynthesis can be severely constrained by low soil moisture[44] and high VPD[43], we suggest that the hot and dry period in 2018 explains why the mean relative increase of $NEE_{800}$ due to warming was not higher under reduced rather than natural herbivory in 2018 (Fig. 4b) as was the case in 2017 (Fig. 4a). In fact, this phenomenon becomes very clear when the effects of warming and reduced herbivory on $CO_2$ fluxes in 2018 are portrayed separately for the period of typical humid conditions (i.e., when VPD was clearly lower than 2.5 kPa) and the period of hydrological stress (Fig. 4c, d). Under typical conditions (Fig. 4c), warming increased $NEE_{800}$ more under reduced rather than normal herbivory, as in 2017, whereas during hydrological stress (Fig. 4d), warming barely elevated $GPP_{800}$ and reduced $NEE_{800}$ under reduced herbivory. Meanwhile, under natural herbivory, the effects of warming on $GPP_{800}$ and $NEE_{800}$ remained positive also during the heat wave (Fig. 4d). These results show that although insect herbivores seem to generally dampen the positive effect of warming on ecosystem $CO_2$ uptake in our Subarctic ecosystem, they can simultaneously preserve $CO_2$ uptake during heat waves due to their cumulative negative effects on plant biomass accumulation. Since plants at high latitudes are poorly adapted to dry conditions[45,46], such unforeseen regulatory factors may be important during the extreme heat waves that are becoming more abundant under climate warming[47].

**Conclusions.** Our results indicate that the generally minor background insect herbivore pressure in high latitude ecosystems can have a strikingly important role in plant resource acquisition and ecosystem–atmosphere $CO_2$ exchange, both in the present and future warmer climates. It also appears that while warming has a clear positive influence on ecosystem $CO_2$ uptake potential, through many simultaneous aboveground and belowground mechanisms, with less herbivory this effect would be greater. As the background insect herbivory is predicted to increase under climate warming[16,25–30,32], these results imply that insects may significantly dampen the strengthening of $CO_2$ sink at high latitudes. A final testimony to the significance of insect herbivores is their ability to influence the way plants and ecosystem $CO_2$ exchange respond to extreme weather patterns, such as the intense heat wave we observed in our study. Altogether, these observations suggest that background insect herbivory should be raised among the key factors in the modeling and empirical research of the responses of high latitude ecosystems to climate warming.

## Methods

**Field site and treatment plots.** The experiment was established in the mountain birch forest-tundra ecotone at the Turku University Kevo Subarctic Research Institute (69°45.4′N, 27°00.5′E; altitude 104 m a.s.l., mean annual temperature −1.3 °C, mean annual precipitation 354 mm in reference years 1981–2010; data from Finnish Meteorological Institute) in 2016. Adult mountain birches were trimmed in the experimental area, but field layer vegetation was left intact and twenty experimental plots (each 0.75 m × 1 m) with no mountain birch stumps were established. To include Subarctic woody species in the experiment and to have controlled and properly replicated plant material to accurately follow treatment responses in plants, plantlets of four birch species (local *B. nana* and *B. pubescens* subsp. *czerepanovii* and currently slightly more southern *B. pendula* and *B. pubescens*) were planted, 20 cm apart, amongst the field layer vegetation in each plot (12 plantlets in each plot, three genotypes per species).

**Plant material and plot properties.** Birch material was collected from natural Subarctic populations growing between 67°43′N and 69°01′N with mean annual temperatures varying from −1.2 to −1.9 °C (reference years 1981–2010; data from Finnish Meteorological Institute). The plantlets were cloned from the collected twig material using the micropropagation technique at the Haapastensyrjä Unit of the Natural Resources Institute Finland (Luke) at the end of 2015, potted into nursery peat (Kekkilä OPM 420 W) in May 2016 and planted to the field site in mid-July 2016. Mean plantlet height was 6.2 cm (SD 4.5, range 1–29 cm) at the time of planting and 15.6 cm (SD 11.0, range 1.5–66 cm) at the end of the growing season 2018. Birches can produce thousands of seeds m⁻² in Subarctic areas[48] and our planting density of 16 plantlets m⁻² is well within the natural variation of density in young birch seedling patches[49]. Birches were chosen as test species due to their high abundance in Subarctic Fennoscandia.

Variation in soil and vegetation properties among the experimental plots was recorded after planting birches. Two to three soil cores (diameter 3 cm, depth 10 cm, or to the nearest rock) were taken from each plot, the samples were pooled, dried (70 °C, 48 h), and the organic matter (OM) content was measured as a loss on ignition (550 °C, 4 h). Lichens and plants were identified to species or genus level and their areal cover was visually estimated for the area inside the flux chamber collar (55 cm × 55 cm). Estimates of soil OM (range 9–38% of soil dry mass among the plots) and the cover of vascular plants (29–83% of area covered; *Empetrum nigrum*, *Vaccinium uliginosum*, and *Vaccinium vitis-idaea* as common species; cover of planted birches was on average 7% of the cover of all vascular plants), mosses (3–85%, *Pleurozium schreberi* and *Hylocomium splendens*) and lichens (0–23%, *Nephroma arcticum* and *Cladonia arbuscula*) were later used as covariates in the statistical models to control the effects of soil and plant plot-to-plot variation on response variables.

**Experimental design and treatments.** Treatments included two levels of warming (ambient, +3 °C) and two levels of insect herbivory (natural, reduced) in a fully factorial 2 × 2 design. In the field, the 20 treatment plots were first divided into five replicate blocks and then in each block, the four treatment combinations were randomly allocated to the plots. The distance between adjacent plots was >1.2 m, which was sufficient to avoid thermal effects between warmed and control plots (ensured using a thermal camera Flir E8; Flir Systems AB, USA).

Warming was realized using two 240 mm × 60 mm ceramic heaters (Elstein-Werk M. Steinmetz GmbH & Co., Germany; dummy heaters in ambient plots), installed 80 cm above the ground and controlled using a microprocessor-based feedback system. The feedback system maintained a fixed temperature differential between the ambient and heated plots using real-time temperature data from Pt100 sensors (Gräff GmbH, Germany) attached to a RMD680 multichannel transmitter (Nokeval Oy, Finland). The sensors were installed 30 cm above the ground (i.e., on the top of the vegetation layer) within green metal plates, mimicking plant leaf surfaces, in five ambient and five heated plots. The ceramic heaters warm surfaces more than the air, so using the temperature of the green metal plates for controlling heating we ensured that vegetation was not overheated. Warming was turned on in spring when the snow-cover melted below 20 cm, and turned off in late autumn when mean daily air temperatures in ambient plots remained permanently below −3 °C. While warming was on, air temperatures were continuously recorded using Pt100 sensors installed under white plastic plates 30 cm above the ground in three control and three warmed plots. Soil temperature and moisture were regularly, but not continuously, measured in 3–5 spots in each plot at the depth of 5 cm using SM150T soil moisture kit (Delta-T Devices Ltd., UK) and Testo 735 thermometer (Testo SE & Co., Germany) attached to a Pt100 sensor. On average, the plates mimicking plant leaves were 3.3 °C, the air 2.3 °C and the soil 1.2 °C warmer in heated than ambient plots across the warming periods in 2016–2018 (in 2016, warming did not cover the entire growing season but was turned on in early July).

The herbivore treatment was started in the beginning of the 2017 growing season and was accomplished by spraying the herbivore reduction plots with 0.1% solution of synthetic pyrethrin (Decis EC25, Bayer Crop-Science, Germany) and the control plots with tap water weekly, using two portable garden sprayers and a protective tent to eliminate wind drift. Decis EC25 has not found to have side-effects on plant growth or chemistry[50] and although deltamethrin, the active ingredient, contains N, the quantities of N that could enter soil during sprayings are negligible. In our experimental plots, the instantaneous top soil (0–10 cm)

mineral N (sum of $NH_4^+$–N and $NO_3^-$–N) availability is, on average, 1.5 µg N per g dry soil (soil sampled in summer 2017). If all N in one spraying of a plot (69 µg N) entered the soil, the instantaneous mineral N availability in the soil would be increased by 0.1%. Similarly, the yearly maximum rate of N addition through sprayings is only 0.1% of the rate (1 g N m⁻² yr⁻¹) that was not found to affect *B. glandulosa* growth in a tundra experiment with comparable shrub vegetation[36].

Warming and herbivore treatments were also carried out during the 2019 growing season, which allowed us to provide supplementary leaf damage data for 2019 (Supplementary Tables 2 and 3).

**Measuring and calculating $CO_2$ fluxes and VPD.** Aluminum collars (outside dimensions 60 cm × 60 cm) were assembled in each plot to enable $CO_2$ exchange measurements with the closed chamber technique with transparent polycarbonate chambers (59 cm × 59 cm × 50 cm and 59 cm × 59 cm × 40 cm)[51]. As the collars were squares and the birch plantlets were planted 20 cm apart in three rows and four columns (following the shape of the plot), nine plantlets (3 × 3) were always included in the collars and three excluded (with the exception of one plot, where only six plantlets were included due to difficulties in the placement of the collar into the rocky soil surface). Each collar had grooves that were filled with moist quartz sand to provide an airtight seal between the collar and the chamber. The $CO_2$ concentration and air temperature inside the chamber were recorded continuously during measurements. A Vaisala CARBOCAP GMP343 (Vaisala Oyj, Finland) and a Picarro G2401 (Picarro Inc., CA, USA) online gas analyzer were used for taking $CO_2$ concentration measurements. The chamber closure time was 6 min for the GMP343 (used during 4 and 7 days in 2017 and 2018, respectively) and 2 min for the G2401 (1 day in 2017, 5 days in 2018). In all cases, the net $CO_2$ exchange was measured under the prevailing light conditions and with a dark hood. To determine the radiation response of photosynthesis, one or two additional shading levels were generated using meshes with 30 and 70% transparency when possible (160 of the 340 cases). The air inside the chamber was mixed with a battery-driven fan.

The solar radiation levels were measured with a photosynthetically active radiation (PAR) sensor (PQS1, Kipp & Zonen) on top of the chamber. In the GMP34₃-based system, there was a lid, made of the chamber material, above the PAR sensor to emulate the conditions inside the chamber, while the sensor was uncovered in the G2401-based system. The influence of the chamber wall on the measured PAR was tested afterwards with a similar transparent chamber, which indicated that the difference between the measurements taken inside and outside the chamber was <2%. In addition, it is important to note that the measurements used in the present study were each day conducted with the same system for all the chamber collars, so no systematic error was introduced into the comparison of treatments.

The $CO_2$ flux, i.e., the net ecosystem exchange (NEE), was calculated as

$$\mathrm{NEE} = \frac{p \times M \times V}{R \times T \times A} \times \frac{\mathrm{dc}}{\mathrm{dt}}, \tag{1}$$

where $p$ is atmospheric pressure, $M$ is the molecular mass of $CO_2$ (44.01 g mol⁻¹), $R$ is the universal gas constant (8.314 J mol⁻¹ K⁻¹), $T$ is the mean air temperature during chamber closure, $V$ is the chamber volume, $A$ is the chamber base area, and $\frac{\mathrm{dc}}{\mathrm{dt}}$ is the mean $CO_2$ mixing ratio change in time calculated with linear regression (see Supplementary Figs. 1 and 2). To allow for the stabilization of the $CO_2$ flux after the chamber closure, the last 5 and 1.5 min of the data series recorded with the GMP343 and G2401, respectively, were included. A micrometeorological sign convention was used: a positive flux indicates a flux from the ecosystem to the atmosphere (emission), and a negative flux indicates a flux from the atmosphere into the ecosystem (uptake).

The measured NEE was partitioned into gross primary productivity (GPP) and ecosystem respiration ($R_e$):

$$\mathrm{NEE} = \mathrm{GPP} + R_e \tag{2}$$

$R_e$ was obtained from the measurements with a darkened chamber. GPP was modeled by a rectangular hyperbola[52]:

$$\mathrm{GPP} = \frac{\mathrm{PAR} \times \alpha \times \mathrm{GP_{max}}}{\mathrm{PAR} \times \alpha + \mathrm{GP_{max}}}, \tag{3}$$

where $a$ is the initial slope between GPP and PAR, and $GP_{max}$ is the theoretical maximum gross photosynthetic rate. The parameters $\alpha$ and $GP_{max}$ were first estimated for those light response measurements that had at least three light levels available and at least one data point with PAR > 800 µmol m⁻² s⁻¹ ($n = 68$ with 20, 15, 16, and 17 measurements from control, warming, herbivory reduction, and warming × herbivory reduction treatment plots, respectively). Using these parameter values, $GP_{max}/\alpha$ ratios were calculated and the median value of 203 µmol m⁻² s⁻¹ was chosen to be used as a common $GP_{max}/\alpha$ ratio. This fixed value was employed in all GPP calculations, while $\alpha$ (or, equally, $GP_{max}$) was available as a free parameter. The reason for introducing a common $GP_{max}/\alpha$ was to obtain consistent data also for those days when irradiance was limited and the saturation level of photosynthesis was not attained. Furthermore, there were no statistically significant differences in $GP_{max}/\alpha$ among the treatment combinations ($P = 0.094$ for the Mood's median test). Using the common $GP_{max}/\alpha$, the $\alpha$ parameter was fitted separately for each light response measurement (i.e., for each chamber collar

in each day, $n = 340$), and $NEE_{800}$ and $GPP_{800}$, i.e., NEE and GPP at PAR = 800 µmol m$^{-2}$ s$^{-1}$, were calculated. $NEE_{800}$ and $GPP_{800}$ represent the $CO_2$ exchange in conditions typical of the daily maximum PAR and serve as standardized metrics that allow comparison of the ecosystem carbon sink potential among different days and treatments. The different steps of the fitting procedure described above are validated in Supplementary Figs. 3–6. We also standardized NEE at a lower (300 µmol m$^{-2}$ s$^{-1}$) and higher (1200 µmol m$^{-2}$ s$^{-1}$) PAR level, but there were no marked differences in the outcome of the statistical analysis (Supplementary Table 8).

The water vapor pressure deficit, VPD (kPa), was calculated as

$$VPD = 0.6107 \times 10^{7.5T/(273.3+T)} \times \left(1 - \frac{RH}{100}\right), \tag{4}$$

where $T$ is air temperature in °C and RH is relative humidity in %.

**Soil N and microbial biomass C.** Plant availability of mineral N ($NH_4^+$ and $NO_3^-$) was estimated in the soil organic layer (at 3 cm depth) using ion-exchange resin bags (UNIBEST Ag Manager™). Three resin bags were placed in each plot in July 2016 and in the following autumns one (2017) or two (2018) bags were transferred from each plot to the laboratory for N extraction in 50 ml of 2 M KCl. The KCl solution was filtered through a glass microfiber filter (Whatman, Germany) and $NH_4^+$ and $NO_3^-$ concentrations were analyzed using a Lachat QuikChem 8000 analyser (Zellweger Analytics, Lachat Instruments Division, USA).

Microbial biomass samples were collected with an auger (diameter 2.5 cm) from three (or more if the soil was shallow) spots within each plot in August 2018. Three layers (organic layer, 0–5 cm layer of mineral soil, >5 cm layer of mineral soil) were separated from each soil core. Within each plot, the samples collected from the same layer were pooled and sieved through a 2-mm (mineral soil) or 6-mm mesh (organic soil) before analyzing microbial biomass carbon (MBC) using the chloroform-fumigation extraction method[53]. In 2018, we further tested whether the insecticide sprayings could have direct effects on soil microbes. Soil from 10 random spots outside the experimental area was collected in early spring, homogenized (largest roots and rocks removed), and placed into ten 1.5-L pots. The pots were buried in the ground and covered with a 1–2 cm layer of dead, oven-dried (70 °C for 20 h, 100 °C for 4 h) *Sphagnum* moss to mimic the insulating moss layer in the experimental plots (where the mean moss depth was 1.2 cm). Along with the sprayings in the experimental plots, five of the pots were then sprayed with water and five with the insecticide. Soil was collected from the pots along with the microbial biomass C sampling and analyzed accordingly. No difference ($t_8 = 0.389$ and $P = 0.708$) between the mean MBC in soils sprayed with water ($546.3 \pm 85.0$ mg MBC kg$^{-1}$ soil; mean ± SE) and insecticide ($494.3 \pm 103.4$) was found (source data are provided as a Source Data file).

**Phenology and growth of *Betula* plantlets.** For estimating treatment effects on plant spring phenology, all birch plantlets were surveyed daily in spring 2017 and 2018. The date of the first bud of a plantlet opening was considered as the start of the growing season for that plantlet. A bud was considered open once the protective bud scales were completely separated and the emerging leaf was visible[54]. During the peak growing season, plant performance was estimated two (2017) to three (2018) times by measuring the chlorophyll content of five full-grown topmost leaves in several branches in each plantlet, using the CCM300 non-destructive optical chlorophyll content meter (Opti-Sciences, USA). The autumn phenology was in turn estimated through chlorophyll breakdown: the chlorophyll content of the five topmost leaves of each plantlet was measured approximately every 3 days using the CCM300 until most of the plantlets had shed their leaves.

To assess plantlet shoot growth, their height was measured at planting and thereafter each autumn following growth cessation. The relative growth was then calculated as a height increment relative to the initial height.

**Leaf herbivore damage of *Betula* plantlets.** To estimate treatment effects on herbivore load, all leaves in each birch plantlet were surveyed for herbivore damage in the middle of August 2017. Leaf damage was illustrated using a Schreiner-type method[20,21,55] in which a damage index, ranging from 0 to 100, is produced for each plant individual by multiplying two scores, $A$ and $B$. The score $A$ is the plantlet mean of values (ranging from 0 to 25) that illustrate the size of the damaged area in a single leave: 0 = no damage, 1 = small damaged area (1–4% of leaf area damaged), 5 = medium damaged area (5–20%) and 25 = large damaged area (>20%). The score $B$ (ranging from 0 to 4) tells the percentage of leaves of a plantlet that are damaged: 0 = 0%, 1 = 1–25%, 2 = 26–50%, 3 = 51–75%, and 4 = 76–100%. As no data was obtained for 2018, we repeated the leaf damage survey in August 2019 to ensure that results from 2017 can be generalized across years.

**Statistical analysis.** Treatment effects on response variables were tested using mixed models and Type I ANOVA (IBM SPSS Statistics 21), where the variance is allocated to explanatory variables in order of their appearance (in tables of statistics, order of explanatory variables follows their order in respective models). Soil OM content and the areal cover of vascular plants, lichens, and mosses were treated as covariates and included in the models first to remove plot-to-plot variation that might otherwise contribute to treatment effects. Warming and herbivory

treatments were treated as fixed effects and field block as a random effect. For birch variables, species was included as a fixed effect and genotype, nested within species, as a random effect. Date, year, and soil layer (MBC was measured for three adjacent soil layers) were treated as repeated measurements and compound symmetry was used as a repeated covariance structure. Those response variables that were measured multiple times during a growing season were tested separately for 2 years. ANOVA assumptions were checked from the residuals and transformations were used where necessary (reported in table and figure legends).

**Reporting summary.** Further information on research design is available in the Nature Research Reporting Summary linked to this article.

## Data availability

The authors declare that the data supporting the findings of this study are available within the paper and its Supplementary Information files. Source data for Figs. 1–3, Supplementary Figs. 1–6, and Supplementary Tables 2 and 8 are provided as a Source Data file.

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

## Acknowledgements

This research was supported by the Academy of Finland (Grants 285030 and 296888) and Kone Foundation (a personal grant to T.S.). The authors thank T. Ryynänen for designing and assembling the heating apparatus, J. Unga and K. Ranta for collecting birch material, T. Lehtosalo, N. Fontaine, T. Dubo, and M. Fontaneu for assisting in data collection, M. Lehtonen for analyzing N, S. Robinson for commenting and editing language, L. Mehtätalo (University of Eastern Finland statistical consulting services) for advising in statistics, and the Kevo staff, especially O. Suominen, I. Syvänperä, and E. Vainio, for providing excellent facilities and assistance throughout the years.

## Author contributions

T.S. and J.M. designed and established the field experiment and coordinated the day-to-day measurements. M.R. helped with site selection and production of birch plantlets together with E.O. $CO_2$ flux measurements were designed by T.S., M.A., and J.M. L.H. and K.M. carried out the measurements and L.H., M.A., and J.-P.T. analyzed the flux data. K.M. carried out phenology surveys, vegetation analysis, and soil moisture and temperature measurements. K.K. and N.M. analyzed microbial biomass. T.S. merged the data, produced tables and figures, and wrote the paper with contribution from all co-authors.

## Competing interests

The authors declare no competing interests.
