## [Peer Review File · Nature Communications]

Reviewers' comments:

Reviewer #1 (Remarks to the Author):

General comments

This is a well written, interesting and important paper that is broadly relevant to understanding ecosystem impacts of climate change in northern ecosystems. The study modified ambient temperature and herbivory rates to compare the individual and combined effects of these two factors on ecosystem C fluxes. The experiment and statistical analyses appear appropriate. My main concern is with about how they recorded CO₂ fluxes and estimated NEE and GPP at a standardized light value. As I understand it, they recorded light responses curves to predict NEE/GPP at a fixed light levels but some unknown portion of the curves recorded never reached the light level required so some kind of assumption was made about these curves which I don't quite understand but that could have implications for absolute values of NEE/GPP and relative differences among treatments, and also makes these kind of analyses difficult to repeat. This issue needs to be clarified and the potential impacts on fluxes/treatment differences needs to be assessed. They also need to present their measured vs fitted GPP values so we can assess how successful the fitting process was (for more detailed information see comments below). Otherwise I have only fairly minor comments, see below.

Specific comments

Lines 16) "tree line forest" makes it sounds like these results are from mature trees, when they are from saplings planted in a forest which is an important and useful distinction. Please clarify here.

Lines 16) "3 oC increase" is referring to the temperature of the actual heating elements, whereas the air and soil warming was lower, correct? If so, revise to use the air or soil warming value, as these are more relevant

Line 18) "90% greater ecosystem CO₂ uptake" could be taken as a statement about the total uptake through the year or growing season, but you measure something rather different. I suggest you stick to the phrasing you use later: ecosystem C uptake potential

Lines 21-22) You miss an opportunity here to specify exactly how/in what direction herbivory may control uptake given (i) your results and (ii) projections of future herbivory in the region

Lines 35-36) "growth losses seem to increase with increasing proportion of leaf damage" this statement seems obvious/trivial...did you mean to say that losses increase nonlinearly with damage or something like this?

Lines 37-38) This isn't really true, there are quite a few studies on herbivore effects on photosynthesis in these systems. See for example references in: Kozlov M.V., Zvereva E.L. (2017) Background Insect Herbivory: Impacts, Patterns and Methodology. In: Cánovas F., Lüttge U., Matyssek R. (eds) Progress in Botany Vol. 79.

Line 41) But I don't think these previous lines of evidence specifically highlight growing season temperature, don't they present annual temperature?

Lines 43-44) "weaken the CO₂ uptake" seems too broad, stick to your wording elsewhere of uptake potential, otherwise people will interpret this as some statement about the total magnitude of the C flux over some fairly long period of time.

Line 46) Specify plot size here. Either here or in the methods you should specify how far were the plots from each other, how were the treatments distributed among the plots?

Line 50) Not very important I guess, but why was it important that the heating apparatus mimic plant leaves? How did they mimic leaves: shape, size, color?

Line 54) Sorry can you clarify what you mean by a 2 by 2 setup

Lines 58-59) Biotic attributes? This seems very vague and mysterious, what are you referring to? Also, it's strange because most of the other variable you have specifically mentioned are also biotic attributes.

Lines 63-65) This seems to be very carefully worded to avoid explicitly stating the absolute rates of herbivore damage observed in your experiment, why? What you say in these lines is also useful/interesting but the absolute rates are surely the most obvious thing to first mention in this section, particularly since you state in various places through the text how low the rates are.

Line 67) "ecosystem C balance"...again this will be taken to mean by many readers as the ecosystem C budget over a longish time scale, but you mean it in a different way. You say "reducing herbivory" but by how much did you reduce it, this will help to interpret the effects on the C fluxes.

Lines 75-76) This statement seems poorly justified, explain a bit more the reasoning why it is unlikely that freely moving herbivores would move towards or away from warming treatments.

Lines 159) I would remove the claim about this being the first study, there have been studies about herbivory, resources acquisition and CO₂, though not all at once necessarily. The results are very interesting and important and speak for themselves.

Line 177) So all of the plots also had shrubs present, not just the birch saplings. Doesn't this make it all the more remarkable that you can detect the effects of herbivory. Why include the shrubs at all if trees were the focus? Do you know if there were any collateral effects of warming and pesticides on the shrubs...though I guess if there was this would have been controlled for statistically?

Lines 194-195) Ah so variation in soil OM and shrubs were was controlled for when assessing changes in C fluxes?

Lines 216-222) Could the pesticide have killed soil organisms, thereby changing soil nutrients and plant growth? See for example:

<https://www.sciencedirect.com/science/article/abs/pii/S0167880997000443>

Lines 224-225) These chambers were smaller than the plots, does this mean that some of the saplings in the plots fell outside the chamber?

Line 226) Did you check what, if any, difference there was between PAR inside and outside of the chamber was? Even transparent chambers usually cut out some PAR.

Lines 229-232) Can you give use some CO₂ accumulation curve metrics under dark conditions (linearity...r²) to assess how good your flux curves/chamber system were. Did you have some system for discarding measured curves or only selecting a portion, or did you use all curves in their entirety to calculate fluxes?? Six minutes is a long time I think, didn't the curves start to saturate after a while? Also, after such a long time, humidity becomes even more of the problem, no? Did you correct you CO₂ fluxes for water vapor dilution (Hooper, D. U. et al. 2002. Corrected calculations for soil and ecosystem measurements of CO₂ flux using the LI-COR 6200 portable photosynthesis system. - *Oecologia* 132: 1-11)? I assume there was a PAR sensor fixed somewhere, was it inside or outside the chamber?

Line 248) Did you consider testing for treatment effects on alpha and GPmax, this would have been very interesting and useful in itself, particularly if your objective was to help models (lines 168 170)

Lines 250-252) How good were these fits (range in RMSE or r² for example). I think it is critical to include in the methods a scatterplot of actual measured GPP and fitted GPP, to get an impression of how well this process worked. You did not make your job easier by selecting such a high PAR value to standardize at, there is a LOT of time in the arctic that is well below that level!

Line 252) More robust than what?

Lines 254-257) This is critical, and as currently described very hard to understand. First, what proportion of all the plot measurements did not fulfil this criteria of having at least one measurement with PAR > 800? Second, in cases where you applied a fit from one plot (with good light conditions) to another (with bad light conditions), was this only done within treatments or was it done across treatments? Or did you estimate a single GPmax/alpha ratio for all of the fits with PAR levels above 800? Can you present us with two scatter plots with the measured vs fitted GPP for (1) the plots that happened to have good light so had their own plot specific GPmax and alpha values and (2) the plots that had bad light so utilized GPmax/alpha ratio from other plots.

Lines 287-289) I like this index but it would be worth presenting each of the components (leaf area damaged, damaged leaves per plant) separately too since the leaf area value would be more easily and directly comparable to previous literature estimates.

Lines 298-299) How were birch species effects analyzed, if I understand right there were the same species in every plot?

Line 440) I'm not sure it is valid to calculate SE from the temporal time series because this treats each repeated measurements per plot as if it they are independents replicate when they are temporally autocorrelated.

Lines 456 - 457) Specify that the lines are solid lines, and remind us which treatments are red and blue/solid and dashed. When you say that soil temperature is represented by dots it is confusing because the dots are also joined by lines, maybe get rid of the lines.

Lines 478-479) Why did you have to fit statistical models to these measurements?

Line 483) Consider using a lighter green that is more easily distinguished from the black.

Reviewer #2 (Remarks to the Author):

This manuscript presents results from a two year field experiment assessing the interactive effects of warming and insect herbivory on ecosystem CO₂ exchange. The study sets out to test the hypotheses that increased levels of background insect herbivory could buffer the increased C sink potential expected under warming. The study reports on the effects of background (non-outbreak) insect herbivory, which is a low intensity but highly prevalent biotic interaction, and warming of air temperatures. The authors report on a comprehensive series of measurements (soil microbial biomass and mineral N availability, leaf chlorophyll, plant growth and phenology) that help support their hypotheses of the underlying mechanisms. It is also interesting that the data cover two growing seasons with contrasting conditions (i.e. heat wave in 2018), and the data available allow making inferences about what influence of such extreme weather conditions will have on ecosystem functioning. The manuscript is nicely and clearly written, and I only have some minor comments, as outlined below.

Specific comments

L15: „more than double“? Recent estimates for tundra herbivory report a 6-7% increase of current values with a 1C increase in temperature (Barrio et al. 2017), whereas previous estimates in boreal forests predicted a 4-5% increase (Wolf et al. 2008)

L35: might be also worth mentioning here the impact that background levels of insect herbivory can have on nutrient cycling in boreal forest ecosystems (Metcalf et al. 2016)

L64: it would be helpful to present here the actual levels of herbivory measured in your study

L73: the unexpected result of warming not increasing leaf damage could be simply because of the short term nature of the experimental manipulations (warming was only applied for 2-3 years). Other studies reporting increased herbivory in warming treatments have a longer time of warming

L73: why would *B. pubescens* have higher levels of herbivory in warmed plots with reduced herbivory?

L76: why is this so? See for example (Moise and Henry 2010)

L122: this is a surprising result, given the short duration of the experiment. For example, in the study by Zvereva et al. (2012) that the authors cite in their paper, it was not until the third year of manipulation of herbivory on saplings of mountain birch (*Betula pubescens* subsp. *czerepanovii*) that differences in shoot length started to become apparent, and only for the highest levels of herbivory (8-16% of leaf area removed). From Suppl Table 2 these rapid responses in growth did not differ between species. Still, it would be good to present exact levels of herbivory per species and how the treatments affected these values.

L191: how were the chambers (55x55 cm) set up in each plot? Were all 12 birch saplings in each plot within the chamber?

L197: nice description of methods! Solved many of the questions I had after reading the brief description in the main text :)

L298: for the „birch“ level models – birch species was significant in several of the models (leaf

damage, leaf chlorophyll content, and involved in interactions in the models for timing of bud break and relative growth). Would it be possible to present the results of models for the different species separately? I am having troubles in seeing the four species pooled together, as they are very different in their life-forms: three trees (two of them outside their native range) and one shrub, and one would expect different levels of herbivory (Kozlov et al. 2015) and responses to experimental warming.

L299: why soil layer?

REFERENCES

Barrio IC, Lindén E, Te Beest M, et al (2017) Background invertebrate herbivory on dwarf birch (*Betula glandulosa-nana* complex) increases with temperature and precipitation across the tundra biome. *Polar Biol* 40:2265–2278. doi: 10.1007/s00300-017-2139-7

Kozlov M V, Filippov BY, Zubrij NA, Zverev V (2015) Abrupt changes in invertebrate herbivory on woody plants at the forest-tundra ecotone. *Polar Biol* 38:967–974

Metcalf DB, Crutsinger GM, Kumordzi BB, Wardle DA (2016) Nutrient fluxes from insect herbivory increase during ecosystem retrogression in boreal forest. *Ecology* 97:124–132

Moise ERD, Henry HAL (2010) Like moths to a street lamp: Exaggerated animal densities in plot-level global change field experiments. *Oikos* 119:791–795. doi: 10.1111/j.1600-0706.2009.18343.x

Wolf A, Kozlov M V, Callaghan T V (2008) Impact of non-outbreak insect damage on vegetation in northern Europe will be greater than expected during a changing climate. *Clim Chang* 87:91–106. doi: 10.1007/s10584-007-9340-6

Zvereva EL, Zverev V, Kozlov M V. (2012) Little strokes fell great oaks: minor but chronic herbivory substantially reduces birch growth. *Oikos* 121:2036–2043. doi: 10.1111/j.1600-0706.2012.20688.x

Reviewer #3 (Remarks to the Author):

The productivity and distribution of plant communities both respond to, and feed back to influence, global carbon fluxes and climate change. Plant productivity and attendant carbon sequestration can also be influenced by insect herbivory. Historically, most emphasis has focused on the impacts of insect outbreaks. More recent research, however, has shown that even low-level, endemic herbivory rates can strongly influence the productivity of plant communities. Very little attention has been directed, however, to how endemic herbivory may alter the impacts of climate warming on carbon exchange in plant communities. This gap in understanding is especially critical for high latitude ecosystems, where climate change is occurring at a rapid rate. The work by Silfver et al. is an important step in that direction.

Silfver et al. employed an experimental site in northern Scandinavia to address how warming (3C) and herbivory (insect removal via insecticide) independently and interactively affect net ecosystem exchange (NEE) of CO₂ in a transitional forest-tundra habitat. The experimental plots were quite small (0.75 m²) and warmed by suspended ceramic lamps. Each plot contained 16 sapling birch (4 reps of 4 species). Numerous plant and soil metrics were monitored over several years. NEE was calculated from physical data, and partitioned into primary production (GPP) and respiration (Re) via standard modelling procedures. Overall, this was a large, complex experiment that appears to have been well-executed, and with data properly analyzed.

Results from the research are both confirmatory and remarkable. Confirmatory in that

experimental warming increased CO₂ uptake (NEE of CO₂ was more negative). Remarkable in that insect herbivory reduced C sink potential by 24-49%. This is an extraordinary impact from a minimal amount of herbivore damage! If this result holds true (more about that anon), this research has very important implications for our understanding of the key factors governing the responses of subarctic ecosystems to climate warming.

The authors also provided compelling explanations for the mechanisms that likely underlie warming- and insect-mediated changes in CO₂ fluxes. The proposed feedback loop incorporating belowground C allocation (fine root growth and exudates), microbial decomposition, and improved soil N availability leading to improved plant growth is very interesting.

Given the central role that insect herbivory occupies in the central storyline, I am surprised that more information was not provided about rates of foliar damage in this experiment. Moreover, I cannot understand the basis for the "damage index" provided. Absent more, and better, data, my enthusiasm for the results of this research remains tempered. Specifically:

1. What were the actual levels of foliar damage? Throughout the manuscript, the authors refer to low levels of endemic herbivory in these habitats, but nowhere do they show what the actual rates of damage on their experimental plants were. Let's see the data. If rates were truly around 1% leaf area (as the authors cite for *Betula* species), then I have difficulty believing how such strong insect responses could be achieved.

2. I don't understand the "Damage Index" provided (Fig. 3). The authors state (lines 285-287) that ALL leaves were monitored (measured?) for damage in 2017. What about 2018? They then go on to state that they calculated a damage index by multiplying the percentage of damaged area on leaves by the percentage of damaged leaves on a plant. This does not make sense to me. Why multiply the two? If all leaves were monitored for damage, and they know how much damage occurred on each leaf, simply provide the mean percent leaf damage (per tree) to readers. Unless the amount of damage per leaf was estimated for just a random subset of leaves on a tree (which does not seem to be the case), there is no need to multiply by percentage of damaged leaves on a tree. (I.e., if I know that insects removed 5% of total leaf area from a tree, why would I then multiply that value by the percentage of leaves to sustain damage on a tree?)

Other suggestions for clarification and improvement:

1. Please provide some additional explanation and rationale for the birch trees used in the experiment. What age and size were the planted saplings. A planting density of 21 trees/m² seems unrealistically high. Can the authors provide some justification? Is this representative of this arctic ecotone? If not, how should the implications of their results be modified?

2. The spectre of "island effects" is always problematic in these types of studies (as I well know!). Can the authors provide any further justification for their declaration (lines 75-76) that abundances of freely moving insects were likely not affected?

3. Vertical growth is not always a good metric of plant productivity, especially when shoots occur in tight spacing. Generally a combination of height and stem diameter is used. Can the authors explain why they did not, and better justify why height is a good measure of productivity?

4. Table 1. This information would be more readily understandable if rows were presented in the standard format of main effects, then interactions, then covariates. Put your interesting experimental results first, not last.

5. The figures are rather "busy", but overall do a good job of illustrating lots of results in a small space. For Fig. 2, I had to study the figures very hard to make out the soil temperature data. Could they be presented differently?

Reviewed by Richard L. Lindroth

Reviewer #1 (Remarks to the Author):

General comments

This is a well written, interesting and important paper that is broadly relevant to understanding ecosystem impacts of climate change in northern ecosystems. The study modified ambient temperature and herbivory rates to compare the individual and combined effects of these two factors on ecosystem C fluxes. The experiment and statistical analyses appear appropriate. My main concern is with about how they recorded CO₂ fluxes and estimated NEE and GPP at a standardized light value. As I understand it, they recorded light responses curves to predict NEE/GPP at a fixed light levels but some unknown portion of the curves recorded never reached the light level required so some kind of assumption was made about these curves which I don't quite understand but that could have implications for absolute values of NEE/GPP and relative differences among treatments, and also makes these kind of analyses difficult to repeat. This issue needs to be clarified and the potential impacts on fluxes/treatment differences needs to be assessed. They also need to present their measured vs fitted GPP values so we can assess how successful the fitting process was (for more detailed information see comments below). Otherwise I have only fairly minor comments, see below.

Our reply 1: We acknowledge that the description of flux calculation methods was incomplete. In the revised version, we have improved the presentation of methods and clarified the issues raised by the reviewer. For details, please see our response to reviewer's specific comments.

Specific comments

Lines 16) "tree line forest" makes it sounds like these results are from mature trees, when they are from saplings planted in a forest which is an important and useful distinction. Please clarify here.

Our reply 2: True, we clarified the experimental set-up better. In addition, 'plantlet', which better describes the planted birches, is now used instead of 'sapling' throughout the ms.

Lines 16) "3 oC increase" is referring to the temperature of the actual heating elements, whereas the air and soil warming was lower, correct? If so, revise to use the air or soil warming value, as these are more relevant

Our reply 3: Revised, we now state both the air and soil warming effects.

Line 18) "90% greater ecosystem CO₂ uptake" could be taken as a statement about the total uptake through the year or growing season, but you measure something rather different. I suggest you stick to the phrasing you use later: ecosystem C uptake potential

Our reply 4: Changed as suggested.

Lines 21-22) You miss an opportunity here to specify exactly how/in what direction herbivory may control uptake given (i) your results and (ii) projections of future herbivory in the region

Our reply 5: We rephrased the ending statement as: "Our results confirm the expected warming-induced increase in high latitude plant growth and CO₂ uptake, but also reveal that herbivorous insects may significantly dampen the strengthening of the CO₂ sink under climate warming".

Lines 35-36) "growth losses seem to increase with increasing proportion of leaf damage" this statement seems obvious/trivial...did you mean to say that losses increase nonlinearly with damage or something like this?

Our reply 6: Although it seems obvious/trivial, this is (to our best knowledge) the only study that has actually showed such linear(ish) response in field conditions. However, we removed this statement as it seems unnecessary.

Lines 37-38) This isn't really true, there are quite a few studies on herbivore effects on photosynthesis in these systems. See for example references in: Kozlov M.V., Zvereva E.L. (2017) Background Insect Herbivory: Impacts, Patterns and Methodology. In: Cánovas F., Lüttge U., Matyssek R. (eds) Progress in Botany Vol. 79.

Our reply 7: We are aware of earlier studies that have measured herbivore effects on leaf chlorophyll fluorescence and photosynthesis, but as far as we know there are no studies that have quantified herbivore effects on ecosystem-atmosphere CO₂ exchange. We revised the text to make this clearer (in lines 40–42).

Line 41) But I don't think these previous lines of evidence specifically highlight growing season temperature, don't they present annual temperature?

Our reply 8: No, in fact, the mean temperature of the warmest summer month (usually July for the northern hemisphere and January for the southern hemisphere) has been used in all latitudinal studies. This is now clarified in the text (line 46).

Lines 43-44) "weaken the CO₂ uptake" seems too broad, stick to your wording elsewhere of uptake potential, otherwise people will interpret this as some statement about the total magnitude of the C flux over some fairly long period of time.

Our reply 9: Changed as suggested.

Line 46) Specify plot size here. Either here or in the methods you should specify how far were the plots from each other, how were the treatments distributed among the plots?

Our reply 10: We added the requested information to the methods (lines 251–255) and the plot size is now given here as well.

Line 50) Not very important I guess, but why was it important that the heating apparatus mimic plant leaves? How did they mimic leaves: shape, size, color?

Our reply 11: When using ceramic or uv-heaters, any surface (leaves, bare soil) warms more than the surrounding air, which is also heavily affected by the wind. Therefore we reasoned that the most reliable way to measure and control the intensity of our experimental warming is to measure the surface temperature of the vegetation. However, as plant leaves move easily with wind (and are often small and/or not pointing to the right direction) and plants can control their leaf temperature, we used "fake leaves" (round green metal plates, 3 cm in diameter) instead to measure surface temperatures and control the heating apparatus. We clarified this reasoning in the methods (lines 263–264).

Line 54) Sorry can you clarify what you mean by a 2 by 2 setup

Our reply 12: Clarified.

Lines 58-59) Biotic attributes? This seems very vague and mysterious, what are you referring to? Also, it's strange because most of the other variables you have specifically mentioned are also biotic attributes.

Our reply 13: Actually, it reads *abiotic* here. However, we added "such as air and soil temperature and moisture" at the end of the sentence to make this clear.

Lines 63-65) This seems to be very carefully worded to avoid explicitly stating the absolute rates of herbivore damage observed in your experiment, why? What you say in these lines is also useful/interesting but the absolute rates are surely the most obvious thing to first mention in this section, particularly since you state in various places through the text how low the rates are.

Our reply 14: Being not explicit was unintentional. We rephrased the text and included absolute rates to better illustrate the level of leaf damage in our study (lines 73–76 and Supplementary Table 2).

Line 67) "ecosystem C balance"...again this will be taken to mean by many readers as the ecosystem C budget over a longish time scale, but you mean it in a different way. You say "reducing herbivory" but by how much did you reduce it, this will help to interpret the effects on the C fluxes.

Our reply 15: "ecosystem C balance" reworded to "ecosystem C uptake potential". The magnitude of herbivory reduction is given in the previous sentence, but we added it here as well ("reducing leaf damages by two thirds") to facilitate a comparison to the magnitude of its effects on the C sink potential.

Lines 75-76) This statement seems poorly justified, explain a bit more the reasoning why it is unlikely that freely moving herbivores would move towards or away from warming treatments.

Our reply 16: We rephrased this part of the text completely (lines 107–114) based on the paper by Moise and Henry (2010). For more discussion about this subject, see also our reply 49.

Lines 159) I would remove the claim about this being the first study, there have been studies about herbivory, resources acquisition and CO₂, though not all at once necessarily. The results are very interesting and important and speak for themselves.

Our reply 17: Removed as suggested.

Line 177) So all of the plots also had shrubs present, not just the birch sapling. Doesn't this make it all the more remarkable that you can detect the effects of herbivory. Why include the shrubs at all if trees were the focus? Do you know if there were any collateral effects of warming and pesticides on the shrubs...though I guess if there was this would have been controlled for statistically?

Lines 194-195) Ah so variation in soil OM and shrubs were was controlled for when assessing changes in C fluxes?

Our reply 18: In quantifying the ecosystem-atmosphere CO₂ exchange, our actual focus was on the entire mountain birch forest ecosystem. However, as it is very difficult to warm and measure the CO₂ exchange of field plots with mature trees, we used plots of intact field layer vegetation complemented with birch plantlets that were carefully planted (by hand) amongst the vegetation. Birch plantlets provided us well controlled (clones of known genotypes of similar age for each species) and properly replicated (3 different genotypes for each species in each plot) plant material by which the warming responses (growth, damage, phenology) could be accurately measured/followed. In the beginning of the experiment, birches accounted for only 7% of the cover of all vascular plants in the chamber collar area. Therefore, large part of the CO₂ exchange was actually due to the shrubs, which apparently benefited (more or less) equally from warming and herbivory removal. We chose 20 as homogenous plots as possible for the experiment, but there was still substantial variation in their soil and vegetation properties (e.g. vascular plant cover ranged between 29–83%). Therefore, we measured the original ("in the beginning") plot-to-plot variation (in soil OM, cover of lichens, mosses and vascular plants including *Betula*) and removed it statistically before assessing the treatment effects on C fluxes (or any other measured variable). We have clarified our experimental set-up and better explained the reasons for using *Betula* plantlets in our study (abstract and lines 54–56, 222–224). See also our replies 48 and 51.

Lines 216-222) Could the pesticide have killed soil organisms, thereby changing soil nutrients and plant growth? See for example:

<https://www.sciencedirect.com/science/article/abs/pii/S0167880997000443>

Our reply 19: In the article above the experimental soil was soaked with 20 times stronger pyrethrin dilution than what we used in our field spraying of plant shoots, thus equally strong harmful effects should not occur. However, we thought about this possibility and therefore tested the direct effects of our insecticide on soil microbes in 2018. The description of the test and it's results - no effects on soil microbial biomass - are now provided in the methods section (lines 355–364).

Lines 224-225) These chambers were smaller than the plots, does this mean that some of the saplings in the plots fell outside the chamber?

Our reply 20: Yes, three plantlets (random species/genotypes) always fell outside the chamber, except for one plot where six plantlets fell out. These details are now better clarified in methods (lines 288–292).

Line 226) Did you check what, if any, difference there was between PAR inside and outside of the chamber was? Even transparent chambers usually cut out some PAR.

Our reply 21: We agree that a transparent chamber wall may cut out part of PAR in comparison to what is measured outside the chamber, but our understanding is that this reduction is very small. The two gas analyzers used in this study had their own chambers. In both systems, the PAR sensor was outside the chamber. In the Vaisala system, there was a lid (made of the chamber material) above the PAR sensor to emulate conditions inside the chamber, while the sensor was uncovered in the Picarro system. The influence of the chamber on the PAR values was not checked with the chambers used in this study, but we tested it afterwards with a similar transparent chamber, which indicated that the difference was < 2%. In addition, it is important to note that the measurements used in the present study were each day conducted with the same system for all the collars, so no systematic error was introduced into the comparison of treatments. These details are now given in the text (lines 303–310).

Lines 229-232) Can you give use some CO₂ accumulation curve metrics under dark conditions (linearity...r²) to assess how good your flux curves/chamber system were.

Our reply 22: We added statistics of the flux determination (least squares linear regression) to Supplementary Information (Supplementary Figure 1). These results demonstrate the high linearity.

Did you have some system for discarding measured curves or only selecting a portion, or did you use all curves in their entirety to calculate fluxes??

Our reply 23: The accumulation curves were generally highly linear, and any nonlinear portion was discarded when the appropriate start and end times were chosen for the data of each chamber closure.

Six minutes is a long time I think, didn't the curves start to saturate after a while? Also, after such a long time, humidity becomes even more of the problem, no? Did you correct you CO₂ fluxes for water vapor dilution (Hooper, D. U. et al. 2002. Corrected calculations for soil and ecosystem measurements of CO₂ flux using the LI-COR 6200 portable photosynthesis system. – *Oecologia* 132: 1–11)? I assume there was a PAR sensor fixed somewhere, was it inside or outside the chamber?

Our reply 24: We did not see any significant signs of saturation in the CO₂ accumulation curves (see reply 22). The Picarro data are internally corrected for water vapour, and they also had a shorter closure time. With the Vaisala GMP343 system, we measured RH inside the chamber, but the sensor broke down during the measurement period. We decided to leave all the data uncorrected for water vapour dilution to ensure a consistent data set. However, we tested the influence of dilution changes on the fluxes during the periods when humidity data were available in 2017. The water vapour

concentration increased in the chamber typically by 0.06 g m^{-3} with the highest increases of 0.178 g m^{-3} during the 6 min closure. The scatterplot of fluxes with and without the dilution correction (Supplementary Figure 2) shows that the effect of water vapour is very limited and thus the dilution correction is not necessary for these data. As highlighted above, the measurements were each day conducted with the same system for all the collars, so no systematic error was introduced into the comparison of treatments. For the PAR sensor, see our reply 21.

Line 248) Did you consider testing for treatment effects on alpha and GPmax, this would have been very interesting and useful in itself, particularly if your objective was to help models (lines 168-170)

Our reply 25: We could not detect statistically significant differences in the GPmax and alpha parameters among the four treatments although GPmax was on average higher in the treatments in which herbivores were reduced.

Lines 250-252) How good were these fits (range in RMSE or r2 for example). I think it is critical to include in the methods a scatterplot of actual measured GPP and fitted GPP, to get an impression of how well this process worked.

Our reply 26: To illustrate the goodness of these fits, we added the following scatterplots to the Supplementary Information.

Supplementary Figure 3: The constant GPmax/alpha ratio used for standardizing the fluxes at $\text{PAR} = 800 \mu\text{mol m}^{-2} \text{s}^{-1}$ was determined as the median from the alpha and GPmax values estimated for the cases where there were three or four light levels available and the highest PAR was over $800 \mu\text{mol m}^{-2} \text{s}^{-1}$. In this figure, we present the measured GPP versus the fitted GPP for the four-level cases ($R^2 = 0.976$ and $\text{RMSE} = 0.0144 \text{ mgCO}_2 \text{ m}^{-2} \text{s}^{-1}$).

Supplementary Figure 4: Using the constant GPmax/alpha ratio (of $203 \mu\text{mol m}^{-2} \text{s}^{-1}$), we fitted the alpha parameter separately for each measurement of the PAR response. The result of these fittings are shown here for the three- and four-point cases ($R^2 = 0.951$ and $\text{RMSE} = 0.0249 \text{ mgCO}_2 \text{ m}^{-2} \text{s}^{-1}$).

Supplementary Figure 5: To illustrate the uncertainty related to extrapolation of the PAR response beyond the highest light level observed, as is necessary in 67% of the cases when standardizing the fluxes at $\text{PAR} = 800 \mu\text{mol m}^{-2} \text{s}^{-1}$, we made an additional test using the data that had at least three PAR levels, the highest of which within $500\text{-}1100 \mu\text{mol m}^{-2} \text{s}^{-1}$. In this test, we removed the data point with the highest light level and fitted the PAR response function to the remaining data. This figure shows the measured GPP at the highest light level against the GPP predicted with this fit ($R^2 = 0.684$ and $\text{RMSE} = 0.0622 \text{ mgCO}_2 \text{ m}^{-2} \text{s}^{-1}$).

You did not make your job easier by selecting such a high PAR value to standardize at, there is a LOT of time in the arctic that is well below that level!

Our reply 27: The reviewer is of course right about arctic light conditions. In the revised version, we replaced the gross photosynthesis in “optimal” radiation conditions GPP1200 (GPP at $\text{PAR} = 1200 \mu\text{mol m}^{-2} \text{s}^{-1}$) by gross photosynthesis in conditions more typical of the daily maximum PAR, i.e. GPP800 (GPP at $\text{PAR} = 800 \mu\text{mol m}^{-2} \text{s}^{-1}$). However, this had a minor influence on the results. We also tested GPP300 (GPP at $\text{PAR} = 300 \mu\text{mol m}^{-2} \text{s}^{-1}$), and even then there were no marked differences in

the outcome of the statistical analysis (only minor effects on NEE, see Supplementary table 8). These comparisons are mentioned in the methods (lines 336–338) with a reference to Supplementary Table 8.

Line 252) More robust than what?

Our reply 28: We meant that GP1200/800 is more robust than GPmax. This part of the text has been reworded.

Lines 254-257) This is critical, and as currently described very hard to understand. First, what proportion of all the plot measurements did not fulfil this criteria of having at least one measurement with PAR > 800? Second, in cases where you applied a fit from one plot (with good light conditions) to another (with bad light conditions), was this only done within treatments or was it done across treatments? Or did you estimate a single GPmax/alpha ratio for all of the fits with PAR levels above 800? Can you present us with two scatter plots with the measured vs fitted GPP for (1) the plots that happened to have good light so had their own plot specific GPmax and alpha values and (2) the plots that had bad light so utilized GPmax/alpha ratio from other plots.

Our reply 29: The description of flux calculation methods has been improved, and the rationale for using a constant GPmax/alpha ratio has been set out. The highest PAR value was below 800 $\mu\text{mol m}^{-2}\text{s}^{-1}$ in 228 of the 340 measurements (17 days, 20 plots). We did not apply the fit of a plot/day to another plot/day as such, but we determined a common alpha-to-GPmax ratio that was applied to all plots and days and fitted the alpha parameter separately for each plot and day. Thus, identical methods and selection criteria were used across the treatments. We clarified the description of methods. For measured vs fitted GPP, see our reply 26.

Lines 287-289) I like this index but it would be worth presenting each of the components (leaf area damaged, damaged leaves per plant) separately too since the leaf area value would be more easily and directly comparable to previous literature estimates.

Our reply 30: We agree and compiled a new table (Supplementary Table 2), which gives all the details used for calculating the leaf damage index. This table also includes results from the year 2019 survey, which we carried out to verify the 2017 results. We also added more original values in the beginning of the results and discussion section.

Line 440) I'm not sure it is valid to calculate SE from the temporal time series because this treats each repeated measurements per plot as if they are independents replicate when they are temporally autocorrelated.

Our reply 31: We have consulted Professor L. Mehtätalo from UEF statistical consulting services about this issue. The date and seasonal treatment means and their SE's were all produced by the fitted statistical model, which has date as a repeated effect and thus takes the temporal autocorrelation into account. Therefore these SE's should be correct.

Lines 456 - 457) Specify that the lines are solid lines, and remind us which treatments are red and blue/solid and dashed. When you say that soil temperature is represented by dots it is confusing because the dots are also joined by lines, maybe get rid of the lines.

Our reply 32: Good points, corrected as suggested.

Lines 478-479) Why did you have to fit statistical models to these measurements?

Our reply 33: For removing the effect of the plot-to-plot variation (represented by the covariates in the statistical models) from all means presented in the graphs (EMMs were also used for illustrating the CO₂ fluxes).

Line 483) Consider using a lighter green that is more easily distinguished from the black.

Our reply 34: Lighter tone is now used for GPP.

Reviewer #2 (Remarks to the Author):

This manuscript presents results from a two year field experiment assessing the interactive effects of warming and insect herbivory on ecosystem CO₂ exchange. The study sets out to test the hypotheses that increased levels of background insect herbivory could buffer the increased C sink potential expected under warming. The study reports on the effects of background (non-outbreak) insect herbivory, which is a low intensity but highly prevalent biotic interaction, and warming of air temperatures. The authors report on a comprehensive series of measurements (soil microbial biomass and mineral N availability, leaf chlorophyll, plant growth and phenology) that help support their hypotheses of the underlying mechanisms. It is also interesting that the data cover two growing seasons with contrasting conditions (i.e. heat wave in 2018), and the data available allow making inferences about what influence of such extreme weather conditions will have on ecosystem functioning. The manuscript is nicely and clearly written, and I only have some minor comments, as outlined below.

Specific comments

L15: „more than double“? Recent estimates for tundra herbivory report a 6-7% increase of current values with a 1C increase in temperature (Barrio et al. 2017), whereas previous estimates in boreal forests predicted a 4-5% increase (Wolf et al. 2008)

Our reply 35: "More than double" refers to Kozlov's 2008 empirical work, where he found that foliar damage in birch forests increased from 1–2% to 5–7% along with a latitudinal gradient from 70°N to 60°N (climatically this change represents roughly the predicted warming in the next 100 years) and was best explained by the parallel increase in summer (July) temperatures. Wolf used this empirical work in her simulations, which (when looking at Table 1 simulations) predict up to 5.5 fold (450%) increase in foliar damage for Scandes and up to 3.1 fold (214%) increase for Russia in the next 100 years (we suppose that the "increase by 4–5%" in Wolf's abstract is not a percentual increase, but

means that the damaged foliar area will increase by 4–5 percentage points within the next 100 years). Barrio’s mean predictions indicate that the mean leaf area affected (currently 1.4%) increases by 0.09 percentage points per 1 °C increase in temperature, which implies that with a 3–4 °C temperature increase (based on moderate climate change scenarios), the foliar damage of tundra dwarf birches could increase 19–26% by the end of this century (using Barrio’s estimates for warmer locations these percentages would be 36–49%). The predictions thus vary greatly depending on location (within the Arctic), tree species and magnitude of temperature change, but also on whether they are based on latitudinal observations or simulations. As this is a too complex matter to touch in the abstract, we reworded the text in the abstract as “may more than double under climate warming” (this statement thus being based on field observations and climate change predictions) and opened the variability among earlier observations and predictions later in the manuscript (lines 93–101).

L35: might be also worth mentioning here the impact that background levels of insect herbivory can have on nutrient cycling in boreal forest ecosystems (Metcalf et al. 2016)

Our reply 36: True, we added this point on line 38.

L64: it would be helpful to present here the actual levels of herbivory measured in your study

Our reply 37: All three reviewers shared this suggestion. We now present the actual leaf damage percentages behind the leaf damage index in the Supplementary Table 2 and also discuss them briefly in the beginning of the results and discussion section (lines 72–76).

L73: the unexpected result of warming not increasing leaf damage could be simply because of the short term nature of the experimental manipulations (warming was only applied for 2-3 years). Other studies reporting increased herbivory in warming treatments have a longer time of warming

Our reply 38: This is one potential explanation, but is not supported by the results of our second leaf damage survey in 2019 when warming still had no effect on damages (i.e. there was no trend of increasing damages). See also our replies 39 and 40 for more discussion about the additional 2019 leaf damage survey and how to interpret plot-level warming experiments.

L73: why would *B. pubescens* have higher levels of herbivory in warmed plots with reduced herbivory?

Our reply 39: This interaction effect is most likely coincidental since we did not find it (or any other warming effect) when we surveyed leaf damages again in August 2019. This reasoning and the new 2019 results are now included in the manuscript (lines 104–106, Supplementary Table 3).

L76: why is this so? See for example (Moise and Henry 2010)

Our reply 40: We expected more leaf damages in warmed plots as field observations and model simulations suggest increasing insect herbivory along warming climate, but did not find evidence of this. However, as Moise and Henry (2010) convincingly argue, plot-level field treatments likely give

biased picture of warming effects on insect herbivores due to the plots being vulnerable to both congregation and avoidance of animals. As a result, they should not be used as an evidence of (no) herbivory responses to climate warming. We added this reasoning to the manuscript (lines 107–114), and also, to better inform the reader of the expected changes in herbivory under warming, we shortly opened the earlier observations and simulations from literature (lines 93–102).

L122: this is a surprising result, given the short duration of the experiment. For example, in the study by Zvereva et al. (2012) that the authors cite in their paper, it was not until the third year of manipulation of herbivory on saplings of mountain birch (*Betula pubescens* subsp. *czerepanovii*) that differences in shoot length started to become apparent, and only for the highest levels of herbivory (8-16% of leaf area removed). From Suppl Table 2 these rapid responses in growth did not differ between species. Still, it would be good to present exact levels of herbivory per species and how the treatments affected these values.

Our reply 41: Zvereva et al. 2012 used simulated herbivory, i.e. mechanically removed different proportions of leaf area. Simulated herbivory often fails to induce those plant responses that are important for biotic interactions (see Hjalten 2004 "Simulating herbivory: Problems and possibilities" in *Insects and Ecosystem Function*, eds. Weisser & Siemann, Ecological Studies, vol 173) and mechanical removal of a small proportion of a plant leaf may not cause effects on plants in the absence of specific elicitors present in insect oral secretions. Mechanical damage can neither simulate plant losses to sap-feeding insects, and lastly, our experimental *Betula* plantlets were smaller and younger than those used by Zvereva et al. and may have had less resources to compensate for the damage. Nevertheless, we agree that GPP responses to herbivory removal are surprisingly strong and added more discussion of factors that could potentially explain the results (lines 154–161). See also our reply 46 that deals with this subject.

The reason we would not like to show detailed, species-specific responses for *Betula* plantlets is that this manuscript is about ecosystem-atmosphere CO₂ exchange and the birch plantlets only account for < 10% of vascular plant cover in our treatment plots (i.e. the "ecosystem"). The mean response of different birch species (including four species with 3 genotypes in each) is used as an estimate for general plant responses and to explain the mechanisms behind changes in CO₂ exchange. We feel that comparing and discussing species-specific responses is beyond this study. See more about the reasons for using *Betula* plantlets as test species in our reply 48. We have clarified our approach and provided better reasoning for it throughout the ms (lines 55–56, 222–224).

L191: how were the chambers (55x55 cm) set up in each plot? Were all 12 birch saplings in each plot within the chamber?

Our reply 42: The plot area was 75 cm x 100 cm and the 12 plantlets were planted 20 cm apart in three rows and four columns. As the chamber collars were squares, nine of the plantlets (random species/genotypes) were included in the collar and three were left out. We clarified this in the methods (lines 288–292).

L197: nice description of methods! Solved many of the questions I had after reading the brief description in the main text :)

Our reply 43: Thank you, good to know that the description was clear and helpful. Yet, some parts of the methods were further improved in response to the reviewers' comments.

L298: for the „birch“ level models – birch species was significant in several of the models (leaf damage, leaf chlorophyll content, and involved in interactions in the models for timing of bud break and relative growth). Would it be possible to present the results of models for the different species separately? I am having troubles in seeing the four species pooled together, as they are very different in their life-forms: three trees (two of them outside their native range) and one shrub, and one would expect different levels of herbivory (Kozlov et al. 2015) and responses to experimental warming.

Our reply 44: The idea of pooling the birch species together is to find a response as general as possible to represent the responses of the vegetation in our treatment plots and to explain the observed treatment effects on CO₂ exchange. Differences among *Betula* species are interesting indeed, but the manuscript would get significantly longer and stray from the main topic (how herbivory and climate warming control ecosystem-atmosphere CO₂ exchange at high latitudes) if we focused on species-specific *Betula* responses. We feel these deserve an entire manuscript to be properly covered. See also our responses 41 and 48 about this matter.

L299: why soil layer?

Our reply 45: MBC was measured from three adjacent soil layers, which were therefore included in the model as repeated measures. This is now clarified in the methods (line 400).

REFERENCES

Barrio IC, Lindén E, Te Beest M, et al (2017) Background invertebrate herbivory on dwarf birch (*Betula glandulosa-nana* complex) increases with temperature and precipitation across the tundra biome. *Polar Biol* 40:2265–2278. doi: 10.1007/s00300-017-2139-7

Kozlov M V, Filippov BY, Zubrij NA, Zverev V (2015) Abrupt changes in invertebrate herbivory on woody plants at the forest-tundra ecotone. *Polar Biol* 38:967–974

Metcalfe DB, Crutsinger GM, Kumordzi BB, Wardle DA (2016) Nutrient fluxes from insect herbivory increase during ecosystem retrogression in boreal forest. *Ecology* 97:124–132

Moise ERD, Henry HAL (2010) Like moths to a street lamp: Exaggerated animal densities in plot-level global change field experiments. *Oikos* 119:791–795. doi: 10.1111/j.1600-0706.2009.18343.x

Wolf A, Kozlov M V, Callaghan T V (2008) Impact of non-outbreak insect damage on vegetation in northern Europe will be greater than expected during a changing climate. *Clim Chang* 87:91–106. doi: 10.1007/s10584-007-9340-6

Zvereva EL, Zverev V, Kozlov M V. (2012) Little strokes fell great oaks: minor but chronic herbivory substantially reduces birch growth. *Oikos* 121:2036–2043. doi: 10.1111/j.1600-0706.2012.20688.x

Reviewer #3 (Remarks to the Author):

The productivity and distribution of plant communities both respond to, and feed back to influence, global carbon fluxes and climate change. Plant productivity and attendant carbon sequestration can also be influenced by insect herbivory. Historically, most emphasis has focused on the impacts of insect outbreaks. More recent research, however, has shown that even low-level, endemic herbivory rates can strongly influence the productivity of plant communities. Very little attention has been directed, however, to how endemic herbivory may alter the impacts of climate warming on carbon exchange in plant communities. This gap in understanding is especially critical for high latitude ecosystems, where climate change is occurring at a rapid rate. The work by Silfver et al. is an important step in that direction.

Silfver et al. employed an experimental site in northern Scandinavia to address how warming (3C) and herbivory (insect removal via insecticide) independently and interactively affect net ecosystem exchange (NEE) of CO₂ in a transitional forest-tundra habitat. The experimental plots were quite small (0.75 m²) and warmed by suspended ceramic lamps. Each plot contained 16 sapling birch (4 reps of 4 species). Numerous plant and soil metrics were monitored over several years. NEE was calculated from physical data, and partitioned into primary production (GPP) and respiration (Re) via standard modelling procedures. Overall, this was a large, complex experiment that appears to have been well-executed, and with data properly analyzed.

Results from the research are both confirmatory and remarkable. Confirmatory in that experimental warming increased CO₂ uptake (NEE of CO₂ was more negative). Remarkable in that insect herbivory reduced C sink potential by 24-49%. This is an extraordinary impact from a minimal amount of herbivore damage! If this result holds true (more about that anon), this research has very important implications for our understanding of the key factors governing the responses of subarctic ecosystems to climate warming.

The authors also provided compelling explanations for the mechanisms that likely underlie warming- and insect-mediated changes in CO₂ fluxes. The proposed feedback loop incorporating belowground C allocation (fine root growth and exudates), microbial decomposition, and improved soil N availability leading to improved plant growth is very interesting.

Given the central role that insect herbivory occupies in the central storyline, I am surprised that more information was not provided about rates of foliar damage in this experiment. Moreover, I cannot understand the basis for the “damage index” provided. Absent more, and better, data, my enthusiasm for the results of this research remains tempered. Specifically:

1. What were the actual levels of foliar damage? Throughout the manuscript, the authors refer to low levels of endemic herbivory in these habitats, but nowhere do they show what the actual rates of

damage on their experimental plants were. Let's see the data. If rates were truly around 1% leaf area (as the authors cite for *Betula* species), then I have difficulty believing how such strong insect responses could be achieved.

Our reply 46: We agree, leaf damage percentages are valuable information for the reader: they are now presented in the text (lines 72–76) and in the Supplementary Table 2. On average, 26% of all *Betula* leaves in our natural herbivory plots were wounded by insects and 83% of these leaves had 1–4% of their leaf area damaged. These values correspond well with earlier estimates of 1–2% of *Betula* leaf area damaged due to background herbivory in these latitudes. We are equally surprised of our strong plant growth response to herbivory reduction. One potential explanation is that the leaf area, where photosynthesis is negatively affected by folivory can be six times larger than the actual consumed area (Zangerl et al. 2002, Nability et al. 2009). In northern deciduous trees, herbivory also induces production of defence compounds, such as phenolics (Nykänen and Koricheva 2004) and when herbivory is reduced, resources allocated to defence can be used for growth. Reducing herbivory possibly also simultaneously reduces leaf pathogens (Hatcher 1995, Biere and Bennet 2013), leading to stronger effect that could be predicted based on the reduction of herbivore damaged area only. Principally, the insecticide could also have direct beneficial effects on plants, but we have tested those earlier and found none. The amount of N that can enter the soil at a spraying is very small in comparison to the available mineral N in soil (only 0.1 % of instantaneously available mineral N) and the yearly rate of N addition through sprayings is also only 0.1% of the rate ($1 \text{ g N m}^{-2} \text{ yr}^{-1}$) that has earlier been found to have **no** effect on *B. glandulosa* growth in tundra (Zamin and Grogan 2012). Further, we tested the effects of the insecticide on soil microbial biomass during the present experiment and found no effects. All this evidence suggests that the response we found in plant growth is not a direct positive effect of the insecticide on plants, but results from herbivory reduction. To open this issue more for the reader, we added the missing parts of above reasoning to the manuscript (lines 156–163, 282–284, and 355–364).

2. I don't understand the "Damage Index" provided (Fig. 3). The authors state (lines 285–287) that ALL leaves were monitored (measured?) for damage in 2017. What about 2018? They then go on to state that they calculated a damage index by multiplying the percentage of damaged area on leaves by the percentage of damaged leaves on a plant. This does not make sense to me. Why multiply the two? If all leaves were monitored for damage, and they know how much damage occurred on each leaf, simply provide the mean percent leaf damage (per tree) to readers. Unless the amount of damage per leaf was estimated for just a random subset of leaves on a tree (which does not seem to be the case), there is no need to multiply by percentage of damaged leaves on a tree. (I.e., if I know that insects removed 5% of total leaf area from a tree, why would I then multiply that value by the percentage of leaves to sustain damage on a tree?)

Our reply 47: The reason for using damage index is that it is feasible and repeatable in field conditions (due to a low number of sufficiently clear damage classes) when a high number of leaves has to be checked and no samples can be collected for estimating the damage in lab conditions (imagine reliably estimating exact leaf damage percentages for tens to over one hundred small leaves for a 5–10 cm tall *B. nana* plantlet surrounded by a much taller vegetation). We have used the same index in our previous studies and found it suitable for revealing various effects from herbivory treatments to

genetic variation. Anyway, you're right, our description was confusing, so we clarified the method description to explicitly state how the index is calculated (lines 377–385 in methods). Damage indices are not available for 2018, so to ensure that 2017 results can be generalized across years, we repeated the leaf damage survey in 2019. This data and the 2019 results are now included in the manuscript (lines 105–106, Supplementary table 3).

Other suggestions for clarification and improvement:

1. Please provide some additional explanation and rationale for the birch trees used in the experiment. What age and size were the planted saplings. A planting density of 21 trees/m² seems unrealistically high. Can the authors provide some justification? Is this representative of this arctic ecotone? If not, how should the implications of their results be modified?

Our reply 48: The rationale for using birch plantlets in the experiment is twofold: to understand the responses of Subarctic woody species to climate warming (*Betula* species are abundant and often dominant all over the forest-tundra ecotone) and to provide controlled plant material for estimating plant responses to warming. Even in the Subarctic, birches can produce up to 20000 seeds m⁻² (Rousi et al. 2019) and birch regeneration often occurs in dense groups in exposed soil patches. Natural densities of over 700 young seedlings m⁻² have been reported for *Betula pubescens* and *B. pendula* (Kinnaird 1974). In this sense, our planting density of 16 plantlets m⁻² (there were 12, not 16 plantlets in each plot) is not unrealistically high although the natural densities of seedlings amidst field layer vegetation (which resembles our case) are lower than in exposed soil patches. With regard to CO₂ fluxes, birch density is not very important as their cover in the beginning of the experiment was only 7% of the cover of all vascular plants in the plots. This reasoning as well as the age and size of the plantlets at the time of planting are now available in methods (lines 222–228, 233–237, 245).

Kinnaird, J. W. Effect of Site Conditions on the Regeneration of Birch (*Betula Pendula* Roth and *B. Pubescens* Ehrh.). *J. Ecol.* **62**, 467-472 (1974).

Rousi M, Possen BJMH, Pulkkinen P, Mikola J (2019) Using long-term data to reveal the geographical variation in timing and quantity of pollen and seed production in silver and pubescent birch in Finland: implications for gene flow, hybridization and responses to climate warming. *Forest Ecology and Management* 438: 25–33.

2. The spectre of “island effects” is always problematic in these types of studies (as I well know!). Can the authors provide any further justification for their declaration (lines 75-76) that abundances of freely moving insects were likely not affected?

Our reply 49: Based on earlier observations of *Betula* leaf damage along latitudes, we expected to find more damage in plantlets growing in warmed plots, but found no significant general effect of warming in 2017, or in 2019 when we repeated the leaf damage survey to ascertain the 2017 results. The reasons for this can be manifold, and as Moise and Henry (2010) argue, treatment plots in warming experiments are always vulnerable to both congregation and avoidance of freely moving animals and may therefore tell little of responses of herbivore abundances under large-scale climatic

changes. We therefore decided to open this issue more, stress that our plot-level responses need to be treated cautiously, and provide a better overview of other predictions of changes in leaf damage based on field observations and simulations (lines 93–114).

3. Vertical growth is not always a good metric of plant productivity, especially when shoots occur in tight spacing. Generally a combination of height and stem diameter is used. Can the authors explain why they did not, and better justify why height is a good measure of productivity?

Our reply 50: Measuring stem diameter was not possible as it could have injured the smallest birch plantlets. However, we counted the leaves and measured the branch lengths of the plantlets, partly because of taking into consideration the different growth forms of the species, but since these measures do not provide any further information of productivity (see Fig & Table 1. below as an example), and to keep the text concise, they are not reported in the manuscript. We also aimed at measuring the productivity of all those vascular plant species that were found (in varying extent) in all plots (i.e. *Empetrum nigrum*, *Vaccinium vitis-idae* and *Vaccinium uliginosum*), but realized that obtaining reliable productivity data from these species was not possible. These reasonings are now better explained in the text (lines 388–391).

Fig. 1. Total stem and branch length of the birch plantlets in ambient (blue symbols) and warming (red) temperature and in plots with reduced herbivory (dotted) and natural herbivory (solid) during 2017–2018.

Table 1. Statistics of warming (ambient and +3 °C) and herbivory (normal and reduced) effects on the total length (main shoot + branches) of the experimental birch plantlets in 2017 and 2018 (N = 232–239 for each year).

	Total length	
	F	P
Initial height 2016	76.2	<0.001
Soil OM content	0.0	0.836
Vascular plant cover	0.2	0.672
Lichen cover	3.4	0.071
Moss cover	6.6	0.012
Year (Y)	156.9	<0.001
Warming (W)	2.7	0.101
Herbivory reduction (H)	8.1	0.005
Betula species (S)	7.1	0.014
W×Y	10.7	0.001
H×Y	7.2	0.007
S×Y	9.1	<0.001
W×H	0.0	0.842
W×S	0.8	0.503
H×S	1.2	0.302
W×H×Y	0.5	0.466
W×S×Y	1.2	0.313
H×S×Y	0.2	0.877
W×H×S	1.4	0.252
W×H×S×Y	0.1	0.972

The data were analyzed using repeated measures linear mixed models and Type I Anova. Year was treated in the model as a repeated measure. Initial height, soil OM content and the areal cover of vascular plants, mosses and lichens were treated as covariates and added to models to remove initial size variation among plantlets and plot-to-plot variation that might otherwise confound the treatment effects (but omitted from the final model if redundant). Field replicate block and birch genotype (nested within species) were included in the models as random effects, but are not reported. Total length was square root transformed. Values of $P < 0.05$ are in bold.

4. Table 1. This information would be more readily understandable if rows were presented in the standard format of main effects, then interactions, then covariates. Put your interesting experimental results first, not last.

Our reply 51: We agree, this is the standard order of explanatory variables. However, as we used Type I Anova, where the variance is allocated to explanatory variables in the order of their appearance in the model, we feel it is important to list the variables in statistics tables in the same order. The “covariates first” order emphasizes that we first removed the effects of soil and plant plot-to-plot variation from CO₂ fluxes and only then tested the treatment effects. This is particularly important with CO₂ fluxes because the plot-to-plot variation was substantial, for instance, in vascular plant cover (29–83% of area covered in different plots). We used blocking and allocated treatments randomly in plots within blocks, but it is still possible that treatment effects partly arise from this variation. The possibility of such bias can be removed using Type I Anova. The reasons for using type I Anova and listing explanatory variables in tables in the same order as in the models is now better explained on lines 393–397.

5. The figures are rather “busy”, but overall do a good job of illustrating lots of results in a small space. For Fig. 2, I had to study the figures very hard to make out the soil temperature data. Could they be presented differently?

Our reply 52: True, it was not easy to tell soil temperature from air temperature as they follow closely each other. We revised the graph: soil temperature mean dots are now presented without connecting lines and should better stand out from the lines of air temperature.

Reviewed by Richard L. Lindroth

Our reply: Thank you for using your valuable time to review our manuscript.

Reviewers' comments:

Reviewer #1 (Remarks to the Author):

As I said before, this topic and experiment is extremely interesting and novel, and the manuscript in general is excellent. The authors have adequately addressed almost all my comments. But I still have serious concerns/questions about their extrapolation approach. If this was just a methodological/analytical detail then I would not be so worried, but I am concerned that it may impact on some of the scientific conclusions. My overarching concern is that their study is trying to assess differences/similarities among different treatments but one of their key measurements – CO₂ fluxes – was partly calculated with a single generic parameter (GP_{max}-alpha ratio) calculated for all treatments. So if I understand this right this would tend to artificially generate similarities across treatments.

Let me try to explain my concerns and see if I misunderstand (certainly possible):

In their responses the authors clarify “The highest PAR value was below 800 $\mu\text{mol m}^{-2}\text{s}^{-1}$ in 228 of the 340 measurements (17 days, 20 plots). We did not apply the fit of a plot/day to another plot/day as such, but we determined a common alpha-to-GP_{max} ratio that was applied to all plots and days and fitted the alpha parameter separately for each plot and day. Thus, identical methods and selection criteria were used across the treatments.”

1) First, a minor confusion, in the manuscript (line 338) you appear to say that you have 68 measurements which reached above 800 PAR but in your response above this number appears to be 112 (340-228), why the discrepancy?

2) So onto the major confusion, to clarify, (i) for 112 (340-228) plot measurements you used specific (to the time and plot) fitted combinations of the alpha and GP_{max} to estimate flux at 800 PAR, then (ii) for the remaining 228 plot measurements you used specific (to the time and plot) fitted alpha BUT a single generic GP_{max} (derived from a single generic alpha to GP_{max} ratio calculated from all of the 112 plot measurements where PAR reached 800). Is this correct? Then (i) I think you need to revise the methods a bit because it is rather unclear that you use specific alpha for all measurements, (ii) the ultimate fluxes at 800 PAR that you estimate will be influenced quite a bit by the subset of measurements (68 or 112 plot measurements?) where you got a direct GP_{max}-alpha estimate. How was this subset distributed across the different treatments? Clearly, if a disproportionate number of them came from a particular treatment then you potentially end up applying this treatment-specific effect across all of the other treatments when you apply the GP_{max} alpha ratio across the majority of measurements which did not make it up to 800 PAR. Alternatively, if there is wide treatment-level variation in GP_{max} but you apply a single mean value to estimate flux at 800 PAR across a large number of measurements across treatments then you end up underestimating treatment differences in standardized flux. Sure enough, you find that warming and herbivory cause similar increases in CO₂ uptake, how much of this is a real biological pattern and how much is caused by the use of the same GP_{max}-alpha ratio use for a large portion of the data? It is not implausible that there could be treatment effects on GP_{max} and therefore the GP_{max}-alpha ratio, indeed you say in response 25 that “GP_{max} was on average higher in the treatments in which herbivores were reduced.”

My suggested steps to address this are: (a) present in the methods or supplementary material the number of measurements per treatment that reached up to 800 PAR, for which you could generate measurement specific alpha and GP_{max}, (b) calculate GP_{max} and GP_{max}-alpha ratios for each treatment from the subset of measurements that reached to 800 PAR (present those too in the supp material) then apply those treatment-specific GP_{max}-alpha ratios to estimate GP_{max} for the corresponding treatment measurements which did not reach up to 800PAR. This way you should be able to end up with standardized flux measurements which are specific to each treatment and have not been “contaminated” by measurements/parameters from the other treatments. Does this make sense?

Reviewer #2 (Remarks to the Author):

Thank you for your work on this revised version of the manuscript. This revision has addressed all my previous comments. I think the manuscript is even clearer now, and it conveys a relevant message. A few very minor things:

L14: maybe clarify here that the main mechanism through which climate warming makes high latitude ecosystems a C sink is through increasing plant production

L36: "and even at their background..." (use "even" to emphasize the contrast with the outbreak densities presented in the previous line?)

L75: this mention of the damage index makes it a bit confusing, since this has not been introduced yet, and the first part of the sentence can be viewed itself in a way as an index of damage (and it is part of the damage index indeed). Maybe it is just a matter of turning around the sentence: "The observed damage index was low (Fig. 3d), as most of the damaged leaves had only 1-4% of their area damaged (Suppl Table 2)".

L110: "... tell little about the responses..."

L245: "cover of planted birches was on average..."

L250: it would be helpful to indicate here when the experimental manipulations started and how long they were going on for. It sounds like the herbivore reductions treatments were still applied in 2019 (Suppl Table 3) but this is not clear from the manuscript

Reviewer #3 (Remarks to the Author):

In this revision of a previous submission, Silfver et al deal with the criticisms and recommendations of reviewers. I liked the earlier version, although I had some concerns about presentation that clouded a clear understanding of the data presented.

Overall, the authors have done a very good, and thorough, job of addressing the reviewer concerns. Indeed, all of my major concerns have been satisfactorily addressed. Kudos to the authors for providing thoughtful and complete answers to the many recommendations and queries from three reviewers!

My comments below are largely for correction/improvement of grammar, typos, etc.:

Line 14: change "supposed" to "anticipated"

Line 14: change "This might" to "This effect might" (In general, avoid using "this" as the subject of a sentence.)

Line 24: change "of CO₂" to "to the CO₂"

Line 36: delete "also"

Line 38: delete "significantly"

Line 66: delete "the"

Line 79: change "damages" to "damage"

Line 94: delete "with"

Line 95: insert "A" prior to "similar"

Line 157: delete the comma

Lines 157-159. For info on how leaf damage can alter photosynthesis in nearby tissue in *Betula*, see Nabity et al. 2012. *Oecologia* 169:905-913.

Line 236: delete "also"

Line 253: change "into" to "to"

Line 388: This explanation seems to be a rather lame excuse for not measuring diameters of plants. Calipers, properly used, can easily measure diameters of small plantlets without damaging them.

Overall, this is a very fine study with significant implications for our understanding of how insects may influence CO₂ exchange. Well done. I look forward to seeing it published.

Reviewers' comments:

Reviewer #1 (Remarks to the Author):

As I said before, this topic and experiment is extremely interesting and novel, and the manuscript in general is excellent. The authors have adequately addressed almost all my comments. But I still have serious concerns/questions about their extrapolation approach. If this was just a methodological/analytical detail then I would not be so worried, but I am concerned that it may impact on some of the scientific conclusions. My overarching concern is that their study is trying to assess differences/similarities among different treatments but one of their key measurements – CO₂ fluxes – was partly calculated with a single generic parameter (GPmax-alpha ratio) calculated for all treatments. So if I understand this right this would tend to artificially generate similarities across treatments.

Let me try to explain my concerns and see if I misunderstand (certainly possible):

In their responses the authors clarify “The highest PAR value was below 800 $\mu\text{mol m}^{-2}\text{s}^{-1}$ in 228 of the 340 measurements (17 days, 20 plots). We did not apply the fit of a plot/day to another plot/day as such, but we determined a common alpha-to-GPmax ratio that was applied to all plots and days and fitted the alpha parameter separately for each plot and day. Thus, identical methods and selection criteria were used across the treatments.”

1) First, a minor confusion, in the manuscript (line 338) you appear to say that you have 68 measurements which reached above 800 PAR but in your response above this number appears to be 112 (340-228), why the discrepancy?

Our reply 1. In fact, 68 is the number of observations that satisfied two separate conditions: (1) it was possible to measure the flux sequentially at three or four light levels, and (2) the highest PAR level was $> 800 \mu\text{mol m}^{-2} \text{s}^{-1}$. In altogether 112 measurements the PAR was $> 800 \mu\text{mol m}^{-2} \text{s}^{-1}$; but in 44 of those, the condition (1) was not fulfilled. We clarified this part of method description in the manuscript (L334–337).

2) So onto the major confusion, to clarify, (i) for 112 (340-228) plot measurements you used specific (to the time and plot) fitted combinations of the alpha and GPmax to estimate flux at 800 PAR, then (ii) for the remaining 228 plot measurements you used specific (to the time and plot) fitted alpha BUT a single generic GPmax (derived from a single generic alpha to GPmax ratio calculated from all of the 112 plot measurements where PAR reached 800). Is this correct?

Our reply 2. This is mostly correct. However, we did not employ two different approaches for different plots. Instead, a single common GPmax/alpha ratio was used in all GPP800 calculations. This value was estimated using the 68 pairs of GPmax and alpha values that were obtained from the fits to

the data as described above in Our reply 1. After calculating the GPmax/alpha ratios, the median ratio was chosen and used in all GPP800 modellings. In contrast, the alpha parameter was fitted separately for each plot in each day ($n = 340$), i.e. all GPP800 and NEE800 values were calculated using a constant GPmax/alpha, but a fully variable alpha. We revised and clarified the description of methods to make this procedure better understandable (L334–349).

Then (i) I think you need to revise the methods a bit because it is rather unclear that you use specific alpha for all measurements, (ii) the ultimate fluxes at 800 PAR that you estimate will be influenced quite a bit by the subset of measurements (68 or 112 plot measurements?) where you got a direct GPmax-alpha estimate. How was this subset distributed across the different treatments? Clearly, if a disproportionate number of them came from a particular treatment then you potentially end up applying this treatment-specific effect across all of the other treatments when you apply the GP-max alpha ratio across the majority of measurements which did not make it up to 800 PAR. Alternatively, if there is wide treatment-level variation in GPmax but you apply a single mean value to estimate flux at 800 PAR across a large number of measurements across treatments then you end up underestimating treatment differences in standardized flux. Sure enough, you find that warming and herbivory cause similar increases in CO₂ uptake, how much of this is a real biological pattern and how much is caused by the use of the same GPmax-alpha ratio use for a large portion of the data? It is not implausible that there could be treatment effects on GPmax and therefore the GPmax-alpha ratio, indeed you say in response 25 that “GPmax was on average higher in the treatments in which herbivores were reduced.”

My suggested steps to address this are: (a) present in the methods or supplementary material the number of measurements per treatment that reached up to 800 PAR, for which you could generate measurement specific alpha and GPmax, (b) calculate GPmax and GPmax-alpha ratios for each treatment from the subset of measurements that reached to 800 PAR (present those too in the supp material) then apply those treatment-specific GPmax-alpha ratios to estimate GPmax for the corresponding treatment measurements which did not reach up to 800PAR. This way you should be able to end up with standardized flux measurements which are specific to each treatment and have not been “contaminated” by measurements/parameters from the other treatments. Does this make sense?

Our reply 3. Yes, this does make sense, and we understand the concern about a potential bias, either due to averaging out or biasing differences among the treatments when a common GPmax/alpha value is used for all data. However, we have reasons to argue that both these concerns are ill-founded. Concerning the averaging effect, Mood’s median test indicates that we cannot conclude that there is a significant difference in the median GPmax/alpha among the four data groups representing different treatments ($p = 0.094$). On the other hand, any possible bias due disproportionate weighting of the groups is inevitably very limited, since the data are distributed almost uniformly across the treatments ($n = 20, 15, 16, 17$ for control, warming, herbivory reduction and warming x herbivory reduction plots, respectively). We believe these arguments justify the use of a common GPmax/alpha

value for the GPP calculations. It is important to note that, even though we fixed the GPmax/alpha parameter, we still had one free parameter (alpha or, equally, GPmax) available to respond to any systematic variability. We included a new figure (Supplementary Figure 6) to demonstrate that the difference between one- vs. two-parameter fits is in fact minor; i.e. a very accurate fit can also be obtained with a fixed GPmax/alpha ratio. All this reasoning is now better described in the text (L334–349).

Reviewer #2 (Remarks to the Author):

Thank you for your work on this revised version of the manuscript. This revision has addressed all my previous comments. I think the manuscript is even clearer now, and it conveys a relevant message. A few very minor things:

L14: maybe clarify here that the main mechanism through which climate warming makes high latitude ecosystems a C sink is through increasing plant production

L36: “and even at their background...” (use “even” to emphasize the contrast with the outbreak densities presented in the previous line?)

L75: this mention of the damage index makes it a bit confusing, since this has not been introduced yet, and the first part of the sentence can be viewed itself in a way as an index of damage (and it is part of the damage index indeed). Maybe it is just a matter of turning around the sentence: “The observed damage index was low (Fig. 3d), as most of the damaged leaves had only 1-4% of their area damaged (Suppl Table 2)”.

L110: “... tell little about the responses...”

L245: “cover of planted birches was on average...”

L250: it would be helpful to indicate here when the experimental manipulations started and how long they were going on for. It sounds like the herbivore reductions treatments were still applied in 2019 (Suppl Table 3) but this is not clear from the manuscript

Our reply 4. All these suggestions are included in the revised manuscript.

Reviewer #3 (Remarks to the Author):

In this revision of a previous submission, Silfver et al deal with the criticisms and recommendations of reviewers. I liked the earlier version, although I had some concerns about presentation that clouded a clear understanding of the data presented.

Overall, the authors have done a very good, and thorough, job of addressing the reviewer concerns. Indeed, all of my major concerns have been satisfactorily addressed. Kudos to the authors for providing thoughtful and complete answers to the many recommendations and queries from three reviewers!

My comments below are largely for correction/improvement of grammar, typos, etc.:

Line 14: change “supposed” to “anticipated”

Line 14: change “This might” to “This effect might” (In general, avoid using “this” as the subject of a sentence.)

Line 24: change “of CO₂” to “to the CO₂”

Line 36: delete “also”

Line 38: delete “significantly”

Line 66: delete “the”

Line 79: change “damages” to “damage”

Line 94: delete “with”

Line 95: insert “A” prior to “similar”

Line 157: delete the comma

Lines 157-159. For info on how leaf damage can alter photosynthesis in nearby tissue in *Betula*, see Nability et al. 2012. *Oecologia* 169:905-913.

Line 236: delete “also”

Line 253: change “into” to “to”

Line 388: This explanation seems to be a rather lame excuse for not measuring diameters of plants. Calipers, properly used, can easily measure diameters of small plantlets without damaging them.

Overall, this is a very fine study with significant implications for our understanding of how insects may influence CO₂ exchange. Well done. I look forward to seeing it published.

Our reply 5. All these suggestions (including a reference to Nability et al. findings) are included in the revised manuscript.

Reviewers' Comments:

Reviewer #1:

None